# Disentangling degradation pathways of narrow bandgap lead-tin perovskite material and photovoltaic devices

Florine M. Rombach, Akash Dasgupta [ID], Manuel Kober-Czerny [ID], Heon Jin [ID], James M. Ball, Joel A. Smith [ID], Michael D. Farrar & Henry J. Snaith [ID] ✉

Narrow bandgap lead-tin perovskites are essential components of next-generation all-perovskite multi-junction solar cells. However, their poor stability under operating conditions hinders successful implementation. In this work, we systematically investigate the underlying mechanisms of this instability under combined heat and light stress (ISOS L-2 conditions) by measuring changes in phase, conductivity, recombination and current-voltage characteristics. We find an increased impact of the redistribution of mobile ions during device operation to be the primary driver of performance loss during stressing, with further losses caused by a slower increase in non-radiative recombination and background hole density. Crucially, the dominant degradation mode changes with different hole transport materials, which we attribute to variations in iodine vacancy generation rates. By quantifying the impact of these mechanisms on device performance, we provide critical insights for improving the operational stability of lead-tin perovskite solar cells.

The ability to rapidly and affordably expand solar power generation capacities will be essential for a successful global transition to renewable energy[1]. Hybrid organic-inorganic metal-halide perovskites are a promising class of emergent photovoltaic materials for this task[2,3], owing to their high absorption coefficient[4], spontaneous exciton dissociation[5], and long-range charge carrier diffusion[6]. Additionally, because their bandgaps are highly tunable by varying the material composition[7], they are well-suited for implementation in multi-junction architectures due to the ability to tune for optimal bandgap combinations. In fact, so-called 'all-perovskite tandem' solar cells show promise due to their potential for higher efficiencies (feasibly over 34% for double junctions or 37% for triple junctions)[8] and promise of lower embodied energy in manufacturing as compared to silicon-based cells[9].

The lowest bandgap (~1.25 eV) perovskites can be achieved by alloying Pb and Sn[10,11], making lead-tin perovskites an ideal material for the low energy absorber in all-perovskite multi-junction solar cells. However, whilst efficiencies of lead-tin perovskite-based devices now exceed 23% in single junction devices[12], 28.5% in all-perovskite double

junctions[13], and 24.3% in triple junctions[14], reports of high stability under operating conditions remain rare. Improving the stability of lead-tin perovskite solar cells is hence essential for the realization of field-deployable all-perovskite multi-junction solar cells.

Previous interventions found to improve the stability of lead-tin perovskite-based solar cells under illumination and heat include the addition of a discontinuous $Al_2O_3$-nanoparticle 'buffer' layer between the perovskite and electron transport layer[15], the controlled increase of polycrystalline perovskite grain size[16] and the removal of the poly(3,4-ethylenedioxythiophene) polystyrene sulfonate (PEDOT:PSS) hole transport layer (HTL)[16,17]. Multiple studies also reported an improvement in the stability and performance of lead-tin perovskite-based devices through the use of protective capping layers that block atmospheric oxygen and moisture, such as sputtered indium-doped tin oxide (ITO)[18], indium-doped zinc oxide[16], or atomic-layer-deposition tin oxide[19]. A few examples of unusually stable lead-tin perovskite devices have been reported[16,19,20], mostly as a component of all-perovskite tandems. Despite these advancements, the consensus in

Department of Physics, University of Oxford, Clarendon Laboratory, Parks Road, Oxford, UK. ✉e-mail: henry.snaith@physics.ox.ac.uk

the field remains that lead-tin PCS tend to have far inferior stability compared to their neat-lead counterparts, even when air exposure is prevented. Significant performance losses are generally observed during the first tens to hundreds of hours of aging under light exposure at elevated temperatures[21].

The inferior stability of lead-tin perovskites as compared to other perovskite compositions is often ascribed to the use of $Sn^{2+}$. The facile oxidation of $Sn^{2+}$ to $Sn^{4+}$ at the perovskite surface upon exposure to air, along with other chemical processes, has been reported to break down the original perovskite phase, form recombination centers, and cause self-p-doping[22,23]. Whilst $Sn^{2+}$ oxidation has been observed to proceed more slowly in the mixed-metal perovskites compared to their neat-tin counterparts[24], surface degradation could significantly affect device performance even if the bulk remains intact[22,25]. Previous investigations by others have also suggested the poor stability of lead-tin PSCs to be due to the formation of a charge extraction barrier, possibly as a result of the formation of degradation products[22], a reaction with PEDOT:PSS at elevated temperatures[16], and/or a conductivity drop at the perovskite grain boundaries[16]. Hence, it remains unclear whether device performance loss over time is dominantly related to the use of the more easily oxidized $Sn^{2+}$ in the perovskite[18,22–24], the different HTL used (commonly PEDOT:PSS)[26], or other factors, and whether these are inherent to the lead-tin perovskite material itself or stem from other aspects of the device.

Herein, we aim to answer these questions by investigating the degradation occurring in various components in the device stack as well as in fully fabricated solar cells. We investigate the evolving properties of lead-tin perovskite films and devices during encapsulated aging under combined light and thermal stressing. We find that changes in morphology and phase purity only become significant after thousands of hours of aging under simulated sunlight at 65 °C. While the absorber layer is structurally stable over hundreds of hours of aging, the corresponding photocurrent generated from solar cells degrades significantly. In devices using PEDOT:PSS HTLs, we largely attribute this rapid device performance degradation to the bias-induced redistribution of mobile ions at interfaces. Whilst the magnitude of this mobile ion-redistribution-induced loss can be greatly reduced by replacing PEDOT:PSS with PTAA, this causes new losses during aging, stemming from increased self-p-doping. We explain the difference in degradation mechanism with varying HTLs by considering the chemical interaction between PEDOT:PSS and the perovskite. By quantifying the impact of each of these factors on device performance over time, we are able to reveal the most relevant degradation pathways for each architecture. Based on this, we make recommendations that enable significant improvements in the stability of lead-tin perovskite solar cells.

## Results

We consider perovskite film deposited both on bare glass and in 'half-stacks' of ITO/HTL/perovskite. Although MA-containing lead-tin perovskites are used in most record efficiency devices, they have been shown to be significantly less thermally stable[17]. In this study, a 'methylammonium (MA)-free' perovskite composition of $FA_{0.83}Cs_{0.17}Pb_{0.5}Sn_{0.5}I_3$ was used, which instead mainly uses formamidinium (FA). These had a photovoltaic bandgap of 1.25 eV (determined by external quantum efficiency (EQE) spectra of fully fabricated devices, Fig. S1 [27]). Films and devices were aged encapsulated at elevated temperatures (65 °C) and illumination in an ambient environment (ISOS L-2 conditions[28]). Throughout this paper, we refer to these stressing conditions as 'aging' for simplicity.

### Structural and morphological changes during aging

We begin by investigating structural changes occurring in the bulk perovskite absorber layers during aging. To characterize crystallographic changes in the films, we performed X-ray diffraction (XRD)

on lead-tin perovskite films on glass or HTLs (Fig. S2). All XRD patterns measured can be fit to a pseudo-cubic perovskite structure, both before and after 2000 h of aging. New XRD peaks emerge in the aged films, which we identify as the orthorhombic yellow $\delta$-$CsSnI_3$ or $\delta$-$CsPbI_3$ phase (Fig. 1a)[29,30]. This phase appears in perovskite films aged on glass (unencapsulated, aged in $N_2$), on ITO/PEDOT:PSS and on ITO/poly(bis(4-phenyl)(2,4,6-trimethylphenyl)amine) (PTAA) half-stacks (encapsulated, aged in ambient conditions) (Fig. S2). In contrast, when perovskite films were not encapsulated and rather exposed directly to ambient conditions, we instead detected $Cs_2SnI_6$ after a few hours, even without exposure to light and heat (Fig. 1a). $Cs_2SnI_6$ has been previously observed by others in lead-tin perovskites exposed to air[31], and can be spontaneously formed from $\delta$-$CsSnI_3$ under ambient conditions at room temperature[32]. Since we do not detect any $Cs_2SnI_6$ in aged, encapsulated samples, we can be confident that our encapsulation successfully inhibits the formation of large amounts of $Sn^{4+}$ within these films during prolonged aging.

We then performed XRD measurements on complete devices (with an ITO/PEDOT:PSS/ $FA_{0.83}Cs_{0.17}Pb_{0.5}Sn_{0.5}I_3$/ethylenediammonium diiodide ($EDAI_2$)/phenyl-C61-butyric acid methyl ester (PCBM)/bathocuproine (BCP)/Cr/Au architecture) which were aged for 627 h, to determine whether the degradation of the perovskite absorber layers in devices is similar to that of films on glass and half-stacks on HTLs (Fig. S2). New peaks appearing at 2θ values of 12.8° and 53.5° can be assigned to $PbI_2$, and other new peaks at 13.2° and 26.5° can be ascribed to either $\delta$-$CsSnI_3$/$\delta$-$CsPbI_3$ or $Cs_2SnI_6$. We also consistently observe a slight expansion in the cubic lattice volume of the main perovskite phase during aging, for films on glass, HTLs, and complete devices (Table S1). This would be consistent with the formation of a slightly more FA- and Pb-rich primary phase whilst $\delta$-$CsSnI_3$ or $Cs_2SnI_6$ evolves from the initial perovskite composition.

Next, we examine the morphological qualities of films of lead-tin perovskite deposited on glass and on HTLs during aging. Fresh films appear smooth and homogenous in both optical microscope (OM) and scanning electron microscope (SEM) images. However, after 2000+ hours of aging, we observe bright yellow regions a few micrometers in size in the OM images (Fig. 1b–g). SEM images reveal that the homogenous polycrystalline form of the perovskite has changed in these regions, taking the form of smaller crystalline domains than the remaining perovskite bulk. These new crystallites start to become visible on the surface of perovskite films after ~1000 h of aging (Fig. S3a–c), although it is possible that small regions begin to form away from the surface of the film before this time. We also observe the emergence of these features in OM images of aged lead-tin perovskite films deposited on ITO/PEDOT:PSS and ITO/PTAA after ~1000 h of aging, where the regions appear larger when PEDOT:PSS is used as the underlying layer (Fig. S3d–k).

To determine the chemical identity of these regions, we perform spatially resolved energy-disperse X-ray scanning electron microscopy (EDX-SEM) imaging (full maps are shown in Fig. S4). EDX indicates that a higher density of Cs and Sn is present within the new crystallites as compared to the rest of the film, whilst the Pb density is similar between the crystallites and the background (Fig. 1h, i). Identifying these regions as $\delta$-$CsSnI_3$ is also consistent with the observations of increased transmittance of light through these regions made by OM, since $\delta$-$CsSnI_3$ has a very wide optical bandgap of 2.55 eV[30], whilst $\delta$-$CsPbI_3$ has a bandgap of 1.7 eV[33]. Hence, there is significant evidence that the observed new crystallites are the $\delta$-$CsSnI_3$ phase detected by XRD measurements.

The new crystallites also contain a slightly increased density of iodine. It is possible that some other degradation products, such as $I_2$, $SnI_2$, or $SnI_4$ also form in these regions, but any crystalline domains of these other possible impurities are too small to be detected by XRD. Additionally, since $\delta$-$CsSnI_3$ has a significantly higher density (4.82 g cm$^{-3}$)[30] than the lead-tin perovskite (3.87 g cm$^{-3}$ for

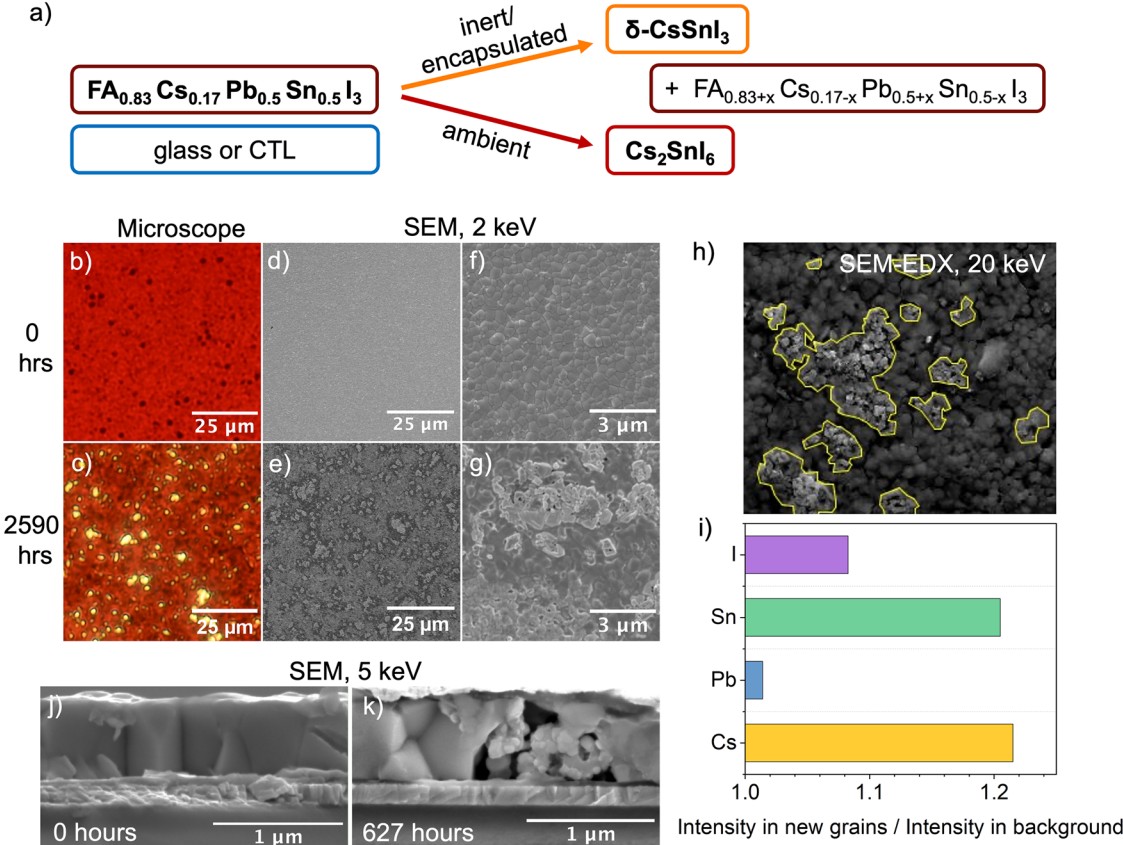

**Fig. 1 | Structural and morphological changes in lead-tin perovskite films during aging. a** Diagram of degradation phase formation observed in $FA_{0.83}Cs_{0.17}Pb_{0.5}Sn_{0.5}I_3$ upon aging in inert and ambient atmospheres. **b**, **c** OM and **d**–**g** SEM images of perovskite films deposited on glass, before and after encapsulated aging under 65 °C and simulated full-spectrum sunlight (76 mW cm$^{-2}$) irradiance for 2590 h. OM images were taken with diascopic illumination, and SEM images were acquired with a 2 kV accelerating voltage. **h** SEM image of an area examined by EDX-SEM analysis, prior to measurement with a 20 keV accelerating voltage. New crystallites are marked with a yellow border. **i** Ratio of mean EDX signal intensity measured within the new crystallites (marked in **h** with a yellow border) to the mean signal intensity in the background (rest of the image). **j** SEM image of a cross-section of the lead-tin perovskite absorber layer inside a fresh device, and **k** SEM image of a device cross-section after encapsulated aging under 65 °C and simulated full-spectrum sunlight (76 mW cm$^{-2}$) irradiance for 627 h, both acquired with a 5 kV accelerating voltage.

$MAPb_{0.54}Sn_{0.46}I_3$)[34], its formation within the film is expected to be accompanied by significant strain leading to void formation. SEM imaging of cross-sections of aged devices confirms this – after 627 h of aging, we observe voids filled with smaller crystallites in the active layer of aged devices, which we don't observe in fresh devices (Fig. 1j, k).

Despite the appearance of these degradation products, the relative XRD peak intensity and width of the original perovskite phase do not change significantly. Although it is possible that some growth of small regions of δ-$CsSnI_3$ may begin immediately during aging, such growth seems sufficiently slow for the perovskite bulk to be left largely intact during 600+ hours of device aging. The measurements lead us to hypothesize that the evolution of the degradation products is relatively slow, and the original lead-tin perovskite phase remains stable over long periods of stressing under elevated temperatures and illumination, even in complete devices. This is corroborated by the fact that the absorption onset of films on glass changes only slightly over hundreds to thousands of hours of aging (Fig. S5) and EQE of aged devices showed no significant change in photovoltaic bandgap during the first ~300 h of aging (Fig. S1b).

## Opto-electronic changes during aging

Having established that the perovskite's bulk structure and optical absorption properties remain stable during the first few hundred hours of aging, we turn our attention to the electronic properties of the film. Smaller-scale changes such as the evolution of point defects can affect background carrier density, mobility, and rates of defect-mediated recombination[31], but may not be captured by the previous methods. We probed changes in the opto-electronic properties of encapsulated lead-tin perovskite films during aging by transient photoconductivity (TPC) and photoluminescence quantum efficiency (PLQE) measurements. We again studied both neat films on glass and films deposited on glass/PEDOT:PSS and glass/PTAA. For samples deposited on HTLs, the crystallinity (from XRD, Fig. S2) and PL peak position (Fig. S6) of films were similar, although small differences in film quality or heterogeneity[35] stemming from the difference in substrate could still contribute to differences in defect formation rates during aging. Our measurements include data from two different batches of glass/perovskite samples, prepared to an identical recipe.

For lead-tin perovskites, both the background hole density ($p_0$) and defect-mediated recombination may increase during aging. The magnitude of $p_0$ is important to device functionality, as it will affect the energetic alignment within devices when it is larger than ~$10^{15}$ cm$^{-3}$ [36], and will impact recombination kinetics when it is comparable to or larger than the photoexcited carrier density. Lead-tin perovskites are slightly p-type, with high-quality films having $p_0$ values of around $10^{14}$ cm$^{-3}$ (previously determined by others using Hall effect measurements[34] and 2-point probe conductivity measurements[37]). Although this lies below the critical densities expected to affect device

performance, it remains unclear to what extent this value may increase during aging.

In this study, we used TPC measurements to determine changes in both $p_O$ and the sum of electron and hole mobilities. In Supplementary Note 1, we discuss assumptions underlying the calculations used to extract these parameters from time-resolved conductivity measurements under pulsed illumination. The method used was further validated by performing TPC measurements at a range of excitation fluences (Fig. S7). An example of the photoconductivity traces from which the sum of mobilities and $p_O$ are derived is shown in Fig. S8. Our resulting estimations of the sum of electron and hole mobility in lead-tin perovskites are shown in Fig. S9a, and are somewhat lower than those previously obtained using THz conductivity[31,38]. This is generally attributed to TPC probing charge transport on a longer length scale, but may also be due to systematic underestimation (discussed further in Supplementary Note 1)[39]. Importantly, we observe that in both neat perovskite and HTL/perovskite samples, the sum of mobilities changes minimally during aging.

We present the estimated $p_O$ of neat perovskite films during aging in Fig. 2a, assuming that electron and hole mobilities are equal (as TPC is only sensitive to their sum). Critically, $p_O$ only begins to increase significantly (and exceed $10^{15}$ cm$^{-3}$) after 150 h of aging. While the true value of $p_O$ may differ from our estimates due to a difference in electron to hole mobility ratio, the observation that the perovskite doping density does not significantly change until 150 h of aging remains valid. This further confirms that the alloying of Sn with Pb successfully avoids

high levels of self-doping/Sn$^{2+}$ vacancy formation, not just immediately after fabrication, but also during at least 150 h of aging.

For HTL/perovskite samples, we were not able to determine an absolute $p_O$ value, owing to challenges in deconvolving the individual contributions of the perovskite layer and the HTL to the total conductance. Instead, we consider relative changes in the absolute dark conductance of the HTL/perovskite stacks in our analysis (Fig. 2b), which should be directly proportional to changes in $p_O$ since mobility was not found to vary significantly. Glass/PEDOT:PSS/perovskite samples show no significant increase in dark conductance during 300 h, whilst glass/PTAA/perovskite stacks exhibit a rapid increase in dark conductance between 100 and 300 h of aging, from ~$1 \times 10^{-8}$ to ~$5 \times 10^{-7}$ S (Fig. S9b). As the conductance of isolated PTAA films is found to not vary during aging (Fig. S9b), we may infer that either the perovskite or the PTAA becomes significantly more conductive by being in contact during aging. For the observed $4 \times 10^{-7}$ S conductance to be mainly due to conduction through the PTAA layer (estimated thickness <10 nm)[40], a conductivity of at least 0.4 S cm$^{-1}$ would be required. This is unlikely to be the case, since the conductivity of PTAA is typically not reported to exceed $10^{-4}$ S cm$^{-1}$ even when extrinsically doped[41]. We hence propose that the observed ~50-fold increase in conductance is dominated by changes in the perovskite, namely due to an increased $p_O$. This increase in p-doping during aging on PTAA seems to occur slightly faster than we observe for films aged on glass. Conversely, aging films on PEDOT:PSS seems to suppress an increase in p-doping. This is further discussed in the context of device aging later in the text.

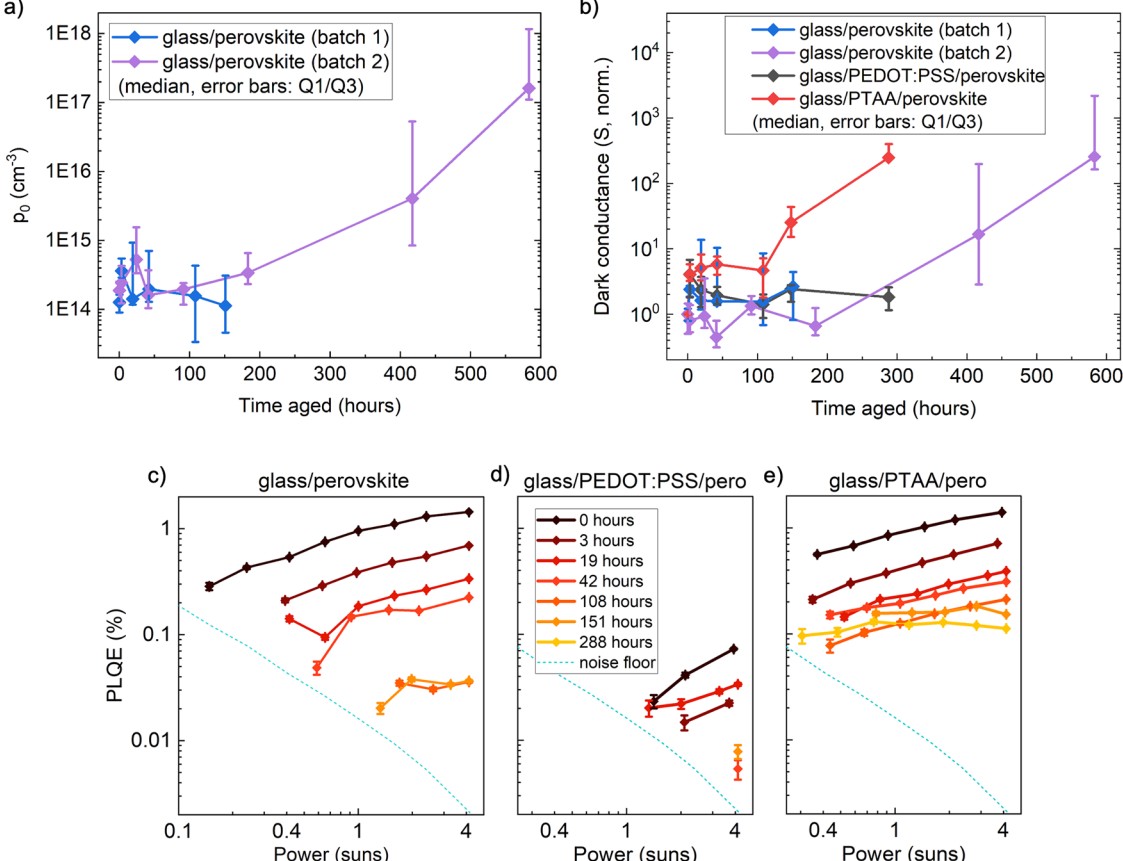

**Fig. 2 | Opto-electronic changes in lead-tin perovskite films during encapsulated aging under 65 °C and simulated full-spectrum sunlight (76 mW cm$^{-2}$) irradiance. a** Background hole density extracted from TPC measurements of lead-tin perovskite films deposited on glass. **b** Dark conductance measured laterally across 300 or 500 μm channels on lead-tin perovskite films, deposited on glass or thin layers of PEDOT:PSS or PTAA. For both (**a**) and (**b**), (n = 4 for batch 1, n = 2 for EDAI$_2$ passivated, n = 3 for batch 3, n = 4 for perovskite on HTL samples). **c** Intensity-dependent PLQE of lead-tin perovskite films deposited on glass (batch 1 in **a**, **b**), **d** glass/PEDOT:PSS or **e** glass/PTAA. Measurements for films on glass were only carried out until 150 h of aging.

Next, we study whether the rate of non-radiative charge carrier recombination changes during aging, by measuring the PLQE of lead-tin perovskite films on glass at excitation fluences ranging from 0.4 to 4 suns equivalent during the first 150 h of aging. PLQE continuously decreases during aging, without much change in the shape of the variation in PLQE across excitation fluences (Fig. 2c). This is consistent with a moderate increase in trap-assisted recombination during aging. Interestingly, this is not accompanied by a significant decrease in long-range mobility, which indicates that any defects formed do not act as effective charge scattering centers. Previous studies using THz conductivity measurements on lead-tin perovskites during air exposure also note an increase in non-radiative recombination without a significant accompanying change in THz mobility[31,38].

For the samples on HTLs, the PLQE of the fresh lead-tin perovskite on PEDOT:PSS is significantly lower than the PLQE of fresh films on glass or PTAA (Fig. 2d). This may be due to the larger hole density in the strongly p-doped PEDOT:PSS, which is expected to increase non-radiative surface recombination at the interface[42]. The PLQE of PEDOT:PSS/perovskite films decreases moderately during aging, falling under the noise limit of our instrument after 19 h. On PTAA, however, the PLQE is initially similar in magnitude and follows a similar trend to that of the perovskite on glass (Fig. 2e). After 151 h, we observe a significant reduction in the gradient of the PLQE against excitation intensity of PTAA/perovskite samples. This is consistent with the increase in background carrier density indicated by TPC - when the background carrier density becomes larger than the steady-state photoexcited carrier density, the rate of radiative recombination tends towards monomolecular rather than bimolecular[43], making the PLQE independent of excitation intensity. Recombination simulations that replicate this change in PLQE with increased p-doping are shown in Supplementary Note 2, Fig. S10, Table S2.

Overall, we find that the $p_O$ and mobility of isolated lead-tin perovskite films on glass are stable during 150 h of aging, with $p_O$ beginning to significantly increase thereafter. Our findings indicate that the presence of HTLs significantly affects the p-doping process, as films on PTAA already show increased p-doping after 100 h, whilst films on PEDOT:PSS do not show increased p-doping even after 288 h of aging. Non-radiative recombination rates increase moderately during aging in all samples. We next relate these effects to changes in device performance during aging.

## Device performance and simulations

For full device performance assessments, we fabricated and encapsulated devices with an architecture of ITO/PEDOT:PSS or PTAA/FA$_{0.83}$Cs$_{0.17}$Pb$_{0.5}$Sn$_{0.5}$I$_3$/EDAI$_2$/PCBM/BCP/Cr/Au and aged these cells under combined thermal and light stressing (65 °C, 0.76 suns equivalent) at open-circuit conditions. We show EQE, light and dark current density-voltage ($J$–$V$) traces in Fig. S11, and maximum power point (MPP) tracking, time-resolved short-circuit current ($J_{sc}$) and open-circuit voltage ($V_{oc}$) in Fig. S12.

In devices using PEDOT:PSS, the MPP tracked power conversion efficiency (PCE) decreases by more than 50% during the first 108 h of aging (Fig. 3a). The $V_{oc}$ is relatively stable during this time, while both the $J_{sc}$ and fill factor (FF) rapidly decay (Fig. 3a). In addition, a notable increase in hysteresis (Fig. S11a) can be observed over time when comparing the reverse (0.9 to −0.2 V) and forward (−0.2 to 0.9 V) $J$–$V$ curve sweeps. The dominance of FF and $J_{sc}$ decay during the degradation of lead-tin perovskite devices with PEDOT:PSS HTLs has been previously observed, even with different lead-tin perovskite compositions[16,22]. In devices using PTAA, initial performance is comparable to that of devices using PEDOT:PSS (Fig. 3a), with significantly less hysteresis (Fig. S11b). Devices using PTAA have a much lower $J_{sc}$ loss during the first 100 h, but continuously degrade and perform worse than devices using PEDOT:PSS after 288 h of aging (Fig. 3a).

For both HTLs used, the $J_{sc}$ expected by integrating the EQE spectrum with the AM1.5 solar spectrum corresponds well to the measured current densities in devices (Fig. S11c, d). We compare the values of the EQE of both devices under high and low wavelength illumination (Fig. 3b). For both HTLs, the difference in EQE at the two wavelengths is relatively small, and the EQE at both is significantly degraded after 300 h of aging. Although we previously showed a slow increase in non-radiative recombination rates during aging, the device measurements here indicate that $J_{sc}$ degradation is dominated by a decrease in charge collection efficiency at one or both CTL interfaces during aging, rather than a decrease in the bulk lifetime of diffusion of charge carriers.

## Impact of mobile ion redistribution on device performance during aging

We postulate that the formation of a charge extraction barrier arises from the redistribution of mobile ions in the perovskite absorber in response to an applied bias. The diffusion of I⁻ vacancies, and possibly also the slower diffusion of other vacancies or interstitials, is possible in metal-halide perovskites with a variety of perovskite compositions[44,45]. In fresh lead-tin PSC devices, the rapid bias-induced redistribution of such mobile ions has previously been shown to cause significant performance losses[37], rationalized to be caused by screening of the bulk electric field across the absorber by the redistributed ions[46,47]. The resulting performance losses also seem to dominate early-time performance degradation for a variety of neat-Pb perovskite compositions[47,48].

The bias-induced redistribution of mobile ions is expected to result in a rapid decay in $J_{sc}$ when measuring the time-resolved current density at short-circuit. In fresh devices with both HTLs, we observe only a small decrease in $J_{sc}$ during the first few seconds of measuring at short-circuit (Fig. 4a), indicating a minor effect of bias-induced mobile ion redistribution. After 108 h of aging, this absolute $J_{sc}$ decay increases to ~11 mA cm$^{-2}$ in devices using PEDOT:PSS, whilst devices using PTAA show a much smaller decay of only ~3 mA cm$^{-2}$. This indicates that the HTL used causes a significant difference in the effects of bias-induced mobile ion redistribution after aging.

To investigate further, we perform $J$–$V$ scans at different scan rates on fresh and aged devices with PEDOT:PSS or PTAA HTLs[49]. Devices are pre-biased close to $V_{oc}$ before carrying out a reverse and forward $J$–$V$ sweep with scan rates varying between 0.2 ms and 7 s scan$^{-1}$. The main bias-driven process expected to affect device performance on this ms-s timescale is the migration of iodine vacancies across the perovskite[44,50–53]. Other mobile ion candidates, such as A- or B-cation vacancies, may still affect device stability by migrating during aging, but as their migration is expected to occur more slowly, its effects are not captured by these measurements. Depending on whether a scan is performed at a slower or faster rate than the characteristic time of ion motion, these measurements reveal $J$–$V$ characteristics of the device with mobile ions either redistributed in response to the applied voltage, or immobile in their pre-biasing position[37]. Comparison of 'slow' and 'fast' $J$–$V$ scans hence reveals how a change in the spatial distribution of the mobile ion (iodine vacancy) population impacts device performance.

We performed such measurements on devices using PEDOT:PSS or PTAA HTLs, and the $J$–$V$ parameters extracted at different scan rates during each aging step are summarized in Figs. S13, S14. This reveals that ion redistribution causes much more severe performance losses in devices using a PEDOT:PSS HTL, both initially and during aging. Examining the shape of the $J$–$V$ curves produced 'slow' (0.18 V s$^{-1}$) and 'fast' (752 V s$^{-1}$) scans (Fig. 4b) shows that in the fast scan, when mobile ions remain at their pre-biasing positions, the large $J_{sc}$ loss previously observed in devices using PEDOT:PSS after aging for 288 h is almost entirely recovered. Although there remains a small FF loss even in the

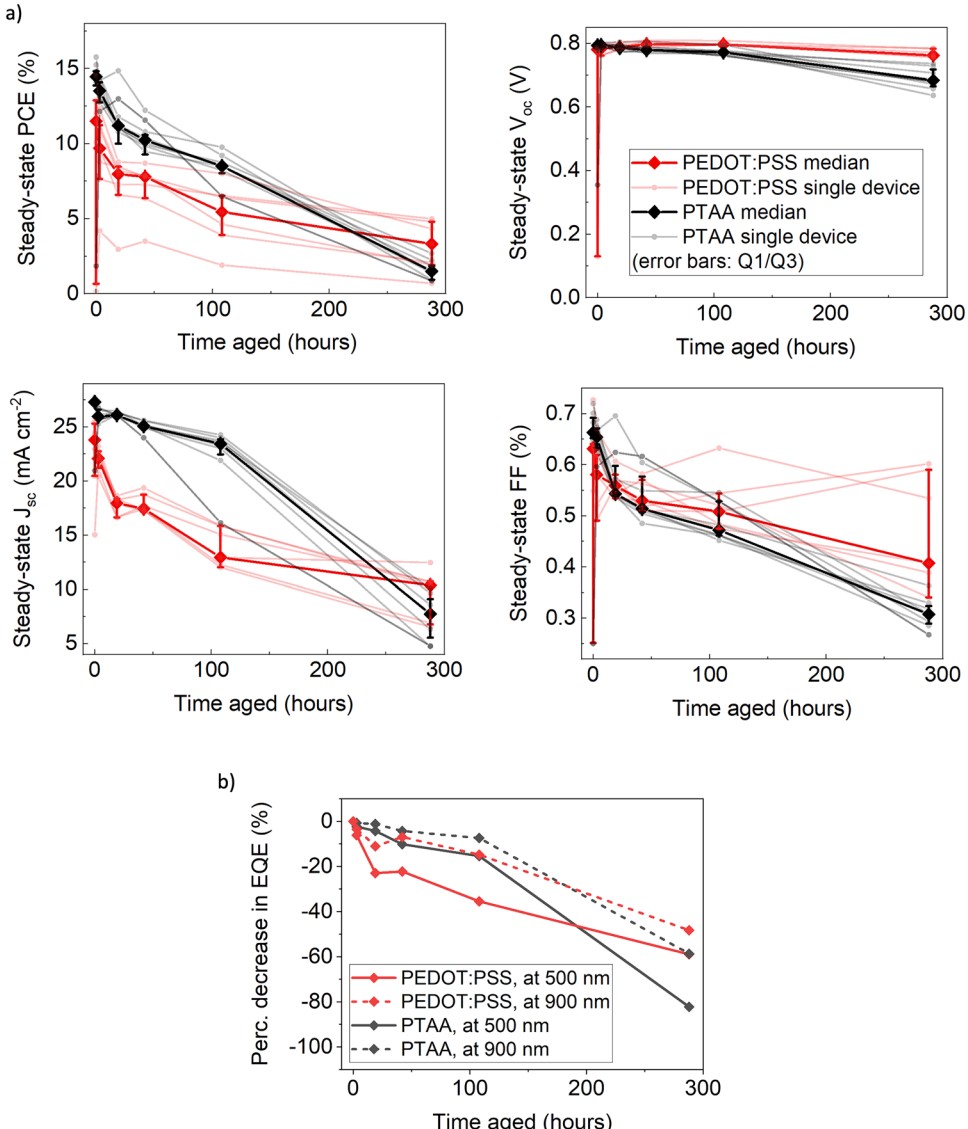

**Fig. 3 | Performance degradation of lead-tin perovskite solar cells during encapsulated aging under 65 °C and simulated full-spectrum sunlight (76 mW cm⁻²) irradiance. a** Performance metrics for encapsulated ITO/HTL/ FA$_{0.83}$Cs$_{0.17}$Pb$_{0.5}$Sn$_{0.5}$I$_3$/PCBM/ BCP/Cr/Au solar cells measured at room temperature under AM1.5 100 mW cm⁻² simulated sunlight at various time intervals during aging using PEDOT:PSS or PTAA HTLs, showing maximum power point power conversion efficiency, steady-state open-circuit voltage, steady-state short-circuit current, and steady-state fill factor of devices during aging. Transparent solid lines represent individual devices, whilst the dark solid line represents the median value with error bars denoting Q1/Q3 ($n = 8$). **b** EQE for devices in a, measured at 500 nm and 900 nm illumination. Full EQE spectra are presented in Fig. S11b.

fast $J–V$ scans, overall performance degradation is minor compared to that observed in the slow scans.

This is strong evidence that the rapid performance degradation observed in lead-tin perovskite devices using PEDOT:PSS is dominated by effects from the bias-induced redistribution of rapidly moving mobile ions. These results stand in contrast to other work, which suggested the possibility of a reduced impact of ion migration in lead-tin perovskites should be lower compared to neat-lead perovskites[54], because of a higher predicted energy barrier to I⁻ vacancy migration with an increased density of Sn$^{2+}$ vacancies[55]. We demonstrate here that this is not the case in lead-tin PSCs with PEDOT:PSS HTLs. This may be because the relatively lower density of Sn$^{2+}$ vacancies in our lead-tin perovskite removes the higher energetic barrier to I⁻ migration, and/or because of a higher density of rapidly moving ions in lead-tin perovskites than previously expected.

In devices using a PTAA HTL, we also observe some difference between fast and slow $J–V$ scans, indicating that mobile ion-induced field screening also contributes to device performance degradation in this case (Fig. 4c). However, we observe significantly more performance degradation in the fast scans in devices aged for 288 h compared to deices using PEDOT:PSS. To compare, in Fig. 4d we plot the difference in the PCE extracted from slow and fast $J–V$ scans during aging for devices with both HTLs, which represents the PCE loss that can be ascribed to the bias-induced redistribution of ions. This clearly shows that for devices using PTAA, the portion of the performance loss that can be ascribed to bias-induced ion redistribution is initially smaller than in devices using PEDOT:PSS, increases minimally during the first 20 h and remains approximately half as large as that of devices using PEDOT:PSS over the entire aging period. In contrast, the losses observed in the fast $J–V$ scan after aging, which must be due to factors

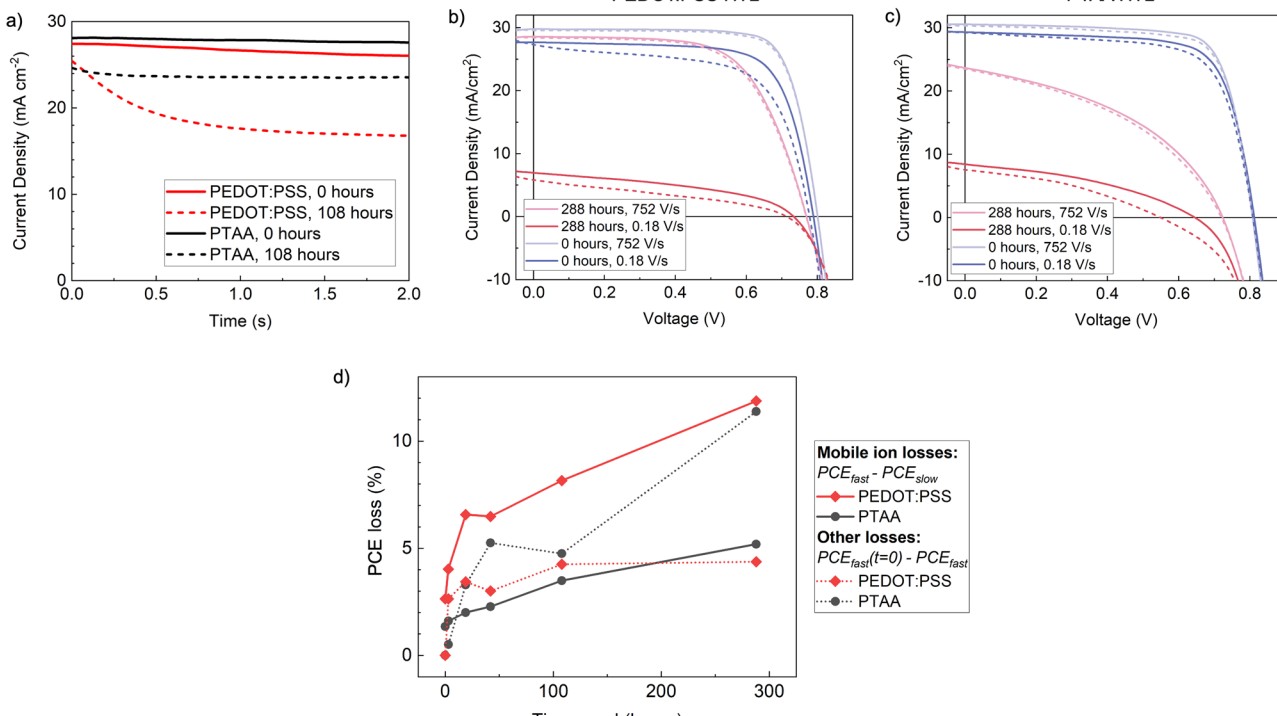

**Fig. 4 | Impact of mobile ions on lead-tin perovskite device performance during encapsulated aging under 65 °C and simulated full-spectrum sunlight (76 mW cm⁻²) irradiance. a** Current density under AM1.5 100 mW cm⁻² simulated sunlight over time at short-circuit of a champion ITO/HTL/FA$_{0.83}$Cs$_{0.17}$Pb$_{0.5}$Sn$_{0.5}$I$_3$/PCBM/BCP/Cr/Au device using a PEDOT:PSS or PTAA/Al$_2$O$_3$ HTL, measured after various periods of aging. **b** Fast 752 V s⁻¹ and slow 0.18 V s⁻¹ J−V scans of a ITO/HTL/FA$_{0.83}$Cs$_{0.17}$Pb$_{0.5}$Sn$_{0.5}$I$_3$/EDAI$_2$/PCBM/BCP/Cr/Au device under AM1.5 100 mW cm⁻² simulated sunlight after 0 and 288 h of aging using a PEDOT:PSS HTL, and **c** the same J−V scans for devices using a PTAA HTL. Solid lines represent the initial reverse scan (decreasing in voltage), and dotted lines represent the subsequent forward scan (increasing in voltage). **d** Plot of the PCE loss extracted from the initial (decreasing in voltage) J−V scans of the champion devices shown in (**a**), after various periods of aging. Losses are divided into 'mobile ion' losses, derived from the difference between fast (752 V s⁻¹) and slow (0.18 V s⁻¹) scans at each aging time, and 'other' losses, derived from the progressive losses observed in the fast (752 V s⁻¹) scans over time.

other than the redistribution of rapidly moving ions, are much worse for devices using PTAA.

**Understanding ion-redistribution-related performance losses**

To understand both the cause of the increased impact of mobile ion redistribution during aging and the observed difference depending on the HTL used, we employ device simulations. Previous work has argued that in neat-Pb perovskites, an increase in mobile ion-related performance losses during aging can best be ascribed to an increase in the density of highly mobile ionic species[47]. However, the impact of the redistribution of mobile ions on device performance can also depend on surface recombination velocities (SRV), and on the properties of the HTL material[56]. To confirm which of these factors best explains our observations, we use the IonMonger drift-diffusion software[56] to simulate J−V curves at the range of scan speeds measured and compare them to the experimental data (parameters used are detailed in Table S3). As Ionmonger cannot take an elevated background hole density into account, we compare the simulations to J−V measurements after 19 h of aging, when we already observe significant performance degradation but do not expect elevated background carrier densities in devices with either HTL.

We simulated the effects of changes in mobile ion density ($N_{ion}$), bulk free carrier lifetime ($\tau_{bulk}$), and SRV on J−V properties of devices with PEDOT:PSS or PTAA HTLs, considering a full range of J−V scan speeds at 1 s - 4 ms per scan. All simulations are shown in the SI (Fig. S15). As expected, both an increase in the density of mobile ions and a decrease in bulk charge carrier lifetime increase the magnitude of ion-redistribution-induced performance losses, for both HTLs. Increasing SRV at the CTL/perovskite interface strongly increased ion-

redistribution-induced losses (mainly in $V_{oc}$) in devices using the highly doped PEDOT:PSS, but only slightly increased them for devices using the lower-doping CTLs PCBM and PTAA,

Next, we compare the simulated J−V parameters to those extracted from device measurements. From a coarse grid search, we select parameters that resulted in the best match of simulations to experimental data and show these in Fig. 5a, b. For devices using PEDOT:PSS, the observed variable-rate J−V data can be well-explained by a combined increase in mobile ion density and bulk recombination rate, whilst for devices using PTAA, increasing the bulk recombination rate alone was sufficient to replicate experimental data. Crucially, for devices with PEDOT:PSS, the experimental data could not be replicated without an increase in mobile ion density. Whilst increasing bulk or surface recombination rates does also increase ion-redistribution-induced losses, simulations with these parameters resulted in significantly different $V_{oc}$ and/or FF than the observed data (Fig. S15b, e). Densities of 10¹⁶–10¹⁷ cm⁻³ iodine vacancies were sufficient to replicate the performance degradation observed, comparable to those found in neat-lead perovskites[57].

The proposed difference in the rate of mobile ion (iodine vacancy) formation in lead-tin perovskites when aged in contact with PEDOT:PSS compared to PTAA can be explained by considering the chemical properties of PEDOT:PSS. PSS contains sulfonic acid groups (-SO$_3$H) which can react with I⁻ from the perovskite to form HI, leaving iodine vacancies in the perovskite film[58]. It is also possible that direct complexation between PEDOT and I⁻ contributes to the production of iodine vacancies in the perovskite[59]. As both iodine vacancies in the perovskite and protons in PEDOT:PSS are mobile (under illumination and/or heat)[3,54], this reaction is not limited to the interface but can

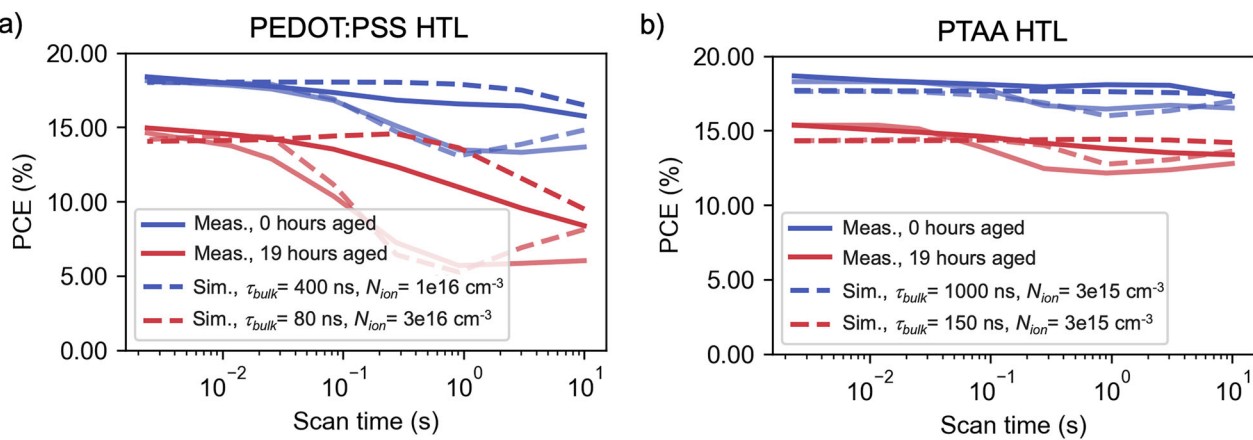

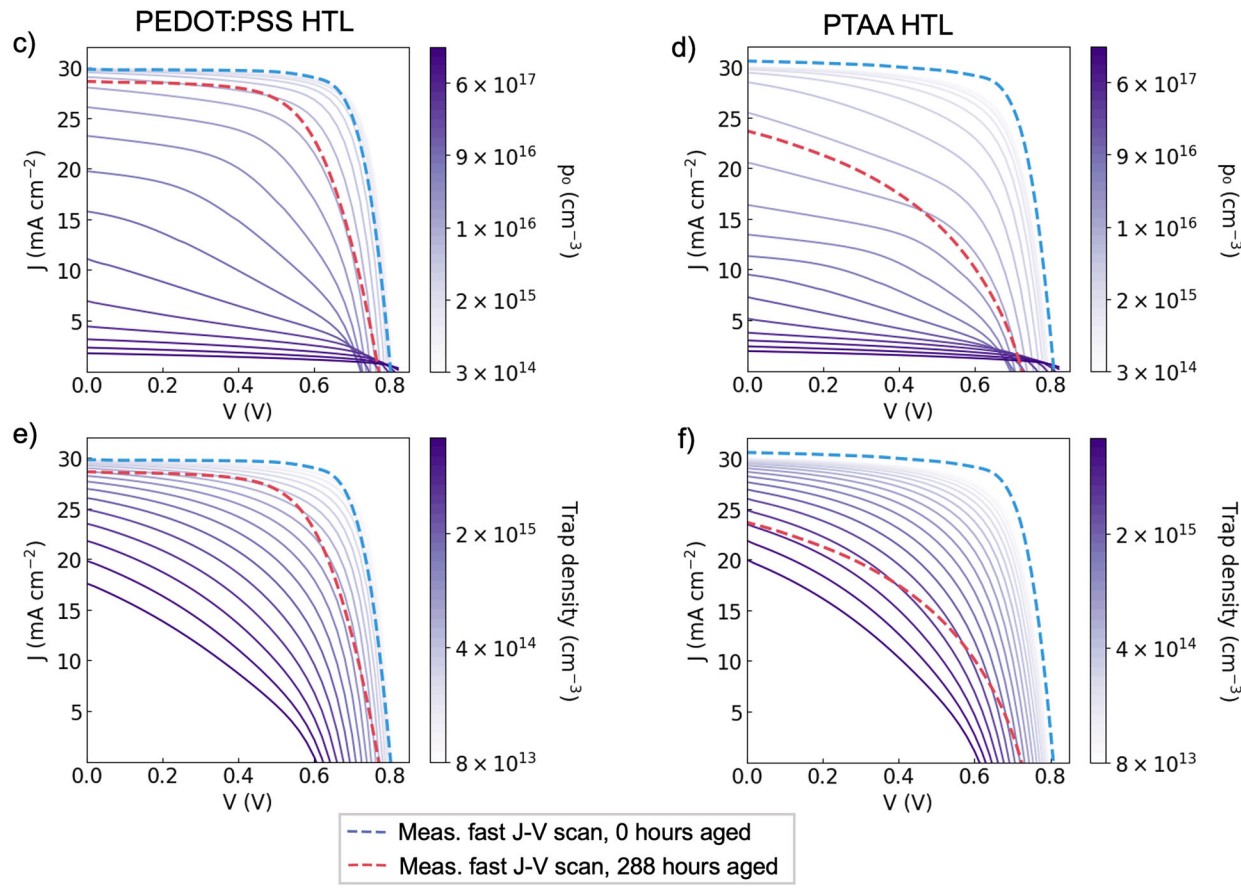

**Fig. 5 | Simulations of lead-tin perovskite device behavior at varying $J$–$V$ scan rates. a** PCE values extracted from backward and forward $J$–$V$ scans with varying scan rates of a lead-tin perovskite device using a PEDOT:PSS HTL and **b** of a device using a PTAA HTL, after 0 and 19 h of encapsulated aging under 65 °C and simulated full-spectrum sunlight (76 mW cm⁻²) irradiance under open-circuit conditions (solid lines). The overlaid dotted lines show PCE values extracted from simulated $J$–$V$ scans, modeled using the Ionmonger software for the same device architectures. Simulations were carried out for a range of perovskite bulk charge carrier lifetimes ($\tau_{bulk}$) and mobile iodine vacancy densities ($N_{ion}$) shown in Fig. S15, with the best fits to experimental data shown here (corresponding parameter values are detailed in the legend). Dark lines represent parameters extracted from the initial reverse scans (decreasing in voltage), and pale lines represent the subsequent forward scan (increasing in voltage). **c** Simulated $J$–$V$ curves modeled using the SCAPS−1D software for a lead-tin perovskite device using a PEDOT:PSS HTL, showing the impact of variation in the background hole density $p_O$ in the perovskite bulk. Dotted lines show the measured $J$–$V$ curves for a device with the same architecture resulting from a very fast (752 V s⁻¹) scan, after 0 and 288 h of encapsulated aging under 65 °C and simulated full-spectrum sunlight (76 mW cm⁻²) irradiance under open-circuit conditions. Equivalent data is show in (**d**) for a device using a PTAA HTL and simulating variation in the background hole density $p_O$ in the perovskite bulk, in (**e**) for a device using a PEDOT:PSS HTL and simulating variation in the deep trap density in the perovskite bulk, and (**f**) for a device using a PTAA HTL and simulating variation in the deep trap density in the perovskite bulk.

continue until the PEDOT:PSS or perovskite layers are significantly depleted. HI vapor is expected to be highly mobile in the perovskite[60] and may easily travel to CTLs or surfaces to react further, such that iodine vacancies are permanently left in the film. In contrast, PTAA is expected to be more chemically inert and is not expected to accelerate the formation of iodine vacancies in these ways. Hence, the increased ion-redistribution-induced performance losses observed during aging when using PEDOT:PSS HTLs are consistent with the formation of a higher density of iodine vacancies due to chemical interaction between PSS and the perovskite.

An often-cited effect of acidic PEDOT:PSS in perovskite solar cells is etching of the ITO electrode layer[61], resulting in $In^{3+}$ migration through the perovskite, which could also be occurring (and affecting interfacial properties) here[62]. However, ion migration affects devices using both PEDOT:PSS and PTAA HTLs at similar scan rates, suggesting the same mobile ion affects both architectures (Fig. S13). We hence deem it unlikely that $In^{3+}$ is responsible for the observed rapidly reversible bias-induced losses.

Given all this, we argue that it is the PEDOT:PSS HTL which causes the observed rapid performance losses in PbSn perovskite solar cells during aging, through an increased population of iodine vacancies. When PTAA is used instead, the portion of device performance losses which can be ascribed to mobile ion redistribution (-5% absolute performance loss after 288 h of aging) is comparable to that of neat-lead devices[47], indicating that the Sn-containing nature of perovskite composition does not seem to significantly impact this performance loss mechanism. Whilst the effects of mobile ions have been much less investigated in neat-tin perovskites, it is possible that the strong effect of the perovskite's high background hole density ($10^{18}$–$10^{20}$ cm$^{-3}$)[63] on the internal distribution of the electric field[36] would outweigh effects from redistribution of a smaller density of mobile ions.

An increase in the density of iodine vacancies in devices using PEDOT:PSS during aging can likely be mitigated not only by using PTAA, but also by using deprotonated PSS[64] or many other suitable chemically inert HTLs. Indeed, treatment of PEDOT:PSS with NaOH has already been observed to significantly increase the stability of lead-tin perovskite solar cells during aging under illumination[26,42]. However, whilst the performance degradation of devices using PTAA is slightly delayed compared to those using PEDOT:PSS, they have overall worse steady-state PCE after 288 h of aging, which must be due to other causes.

### Impact of increased p-doping on device performance during aging

Devices using PTAA experience severe performance losses after 100+ hours of aging, which remain present even in 'fast' (752 V s$^{-1}$) $J$–$V$ scans. These must hence be due to causes other than an increased impact of mobile ion redistribution. We previously inferred a ~50-fold increase in the background hole density of the perovskite absorber after 288 h of aging in contact with PTAA, which could affect device performance. To quantify how this and other changes in material properties affect device performance without the redistribution of mobile ions, we performed drift-diffusion simulations using the SCAPS-1D simulation package (parameters used are detailed in Table S4). Unlike the previously used Ionmonger package, the SCAPS-1D simulation can account for doping in the perovskite layer, but does not simulate ion redistribution[65]. The background hole density $p_O$ and deep trap density in the perovskite absorber layer are presented in Fig. 5b–f as a function of parameter values. Changes in the shallow trap density in the perovskite, trap density at the HTL/perovskite and perovskite/ETL interfaces, and changes in HTL doping density and mobility were also simulated and are presented in Fig. S16.

We compare simulated $J$–$V$ curves to measured fast $J$–$V$ scans of champion devices, which reflect device performance with ions immobile at pre-biasing positions (Fig. 5b–f). Our comparisons show

that in devices using PTAA, the severe PCE losses observed after 288 h of aging can be best reproduced by increasing the $p_O$ of the lead-tin perovskite to ~$10^{16}$ cm$^{-3}$ (Fig. 5d), consistent with our previous TPC measurements. In contrast, for devices using PEDOT:PSS, the much smaller change observed in the fast $J$–$V$ scans after 288 h of aging can be accounted for by a slight increase in deep trap density and/or $p_O$, consistent with our earlier simulations and PL measurements (Fig. 5c, e). Hence, we conclude that the non-mobile ion-redistribution-related performance degradation observed in devices using PTAA is dominantly due to a large increase in the $p_O$ of the perovskite, and that this degradation mode does not occur in devices using PEDOT:PSS.

This large increase in background hole density during aging is not likely to be due to chemical interaction between PTAA and the perovskite, which is relatively inert and has been used to fabricate stable neat-Pb perovskite solar cells[66]. We instead propose that some process p-dopes the perovskite during aging independently of the HTL used, but that the previously described chemical interaction between the perovskite and PEDOT:PSS counteracts this p-doping (Fig. 6). Significant iodine vacancy formation is expected to n-dope perovskites, or shift the Fermi level towards vacuum[53]. Hence, a p-doping process (such as $Sn^{2+}$ vacancy formation) would be counteracted by increased iodine vacancy formation. This mechanism has been previously experimentally observed in CsSnI$_3$[67]. Additionally, the HI produced by the reaction between acidic PEDOT:PSS and the perovskite could further act as a reducing agent to counteract p-doping processes. This explanation is consistent with both the increased mobile ion-redistribution-related performance losses observed in devices using PEDOT:PSS, and the increased $p_O$ observed when perovskite is aged on PTAA.

This indicates that whilst reducing the rate of iodine vacancy formation in lead-tin perovskite devices will reduce performance losses related to the redistribution of mobile ions, doing so will also allow p-doping-related losses to increase (as demonstrated in devices using PTAA). Ultimately, both iodine vacancy formation and p-doping must be avoided to achieve required stabilities in lead-tin perovskite solar cells. Whilst we have demonstrated that the increase in background hole density $p_O$ is relatively slow in encapsulated lead-tin perovskite films and devices during aging (compared to devices exposed to air, or neat Sn perovskites), increased p-doping becomes a performance-limiting factor after -100 h of aging under illumination and heat for the devices using a PTAA HTL. Further research is necessary to determine how to stop this self-doping process, clarifying whether it can be ascribed to the slow ingress of oxygen (and can hence be avoided by even better encapsulation), or to some other chemical interaction within the device (for example, with residual DMSO[68,69]).

### Discussion

In conclusion, we have revealed several different degradation modes affecting the performance of lead-tin perovskite solar cells during encapsulated aging under 65 °C temperatures and 1 sun equivalent illumination. Significant losses due to an increased density of mobile ions occur within days, those due to increased non-radiative recombination and p-doping occur after weeks, and the formation of a CsSnI$_3$ phase occurs after months.

An increased impact of the redistribution of mobile ions is the primary performance degradation mode in devices using PEDOT:PSS. This is ascribed to a rapid increase in the density of iodine vacancies in the perovskite during aging, occurring due to chemical interaction between the perovskite and acidic PEDOT:PSS. We hence establish that avoiding charge transport materials that readily react with iodine in the perovskite is essential to improving the stability of lead-tin PSCs. When PEDOT:PSS is replaced by PTAA, performance losses due to the redistribution of mobile ions become comparable to those observed in neat-Pb perovskite solar cells. To completely eliminate these losses, a better understanding of iodine chemistry within the entire device may

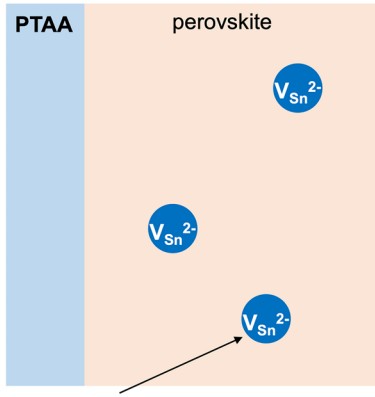
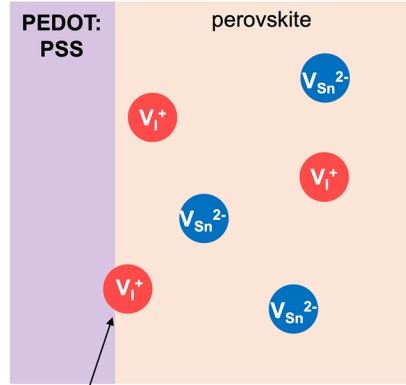

**a) Increased Sn²⁺ vacancies only**

> Increased $p_0$
> Constant mobile ion density

Sn²⁺ vacancies produced slowly

**b) Increased Sn²⁺ vacancies & I⁻ vacancies**

> Low $p_0$
> Increased mobile ion density

I⁻ vacancies produced rapidly, by PEDOT:PSS-perovskite reaction

**Fig. 6 | Diagram illustrating a mechanism consistent with the observed differences in the behavior of lead-tin perovskite solar cells with PTAA or PEDOT:PSS HTLs after aging. a** Devices using PTAA are proposed to mainly experience an increase in low mobility p-doping defects (such as the Sn2+ vacancies shown here) during aging, resulting in an increased background hole density $p_O$ but a constant mobile ion density. **b** Devices using PEDOT:PSS are proposed to additionally experience an increase in mobile I⁻ vacancy density. This counteracts the effect of the p-doping defects on the Fermi level, keeping the background hole density $p_O$ low.

help develop strategies to further reduce iodine vacancy formation during both fabrication and aging.

Lead-tin PSCs employing PTAA are less affected by mobile ion-redistribution-related losses during aging, likely due to reduced chemical interaction between the PTAA and iodine in the perovskite. However, an increase in p-doping during aging leads to even worse overall performance degradation in devices using PEDOT:PSS after a few hundred hours. We suggest that this increased p-doping is not caused by PTAA, but rather a process inherent to the lead-tin perovskite material, which is counteracted by the formation of iodine vacancies in devices using PEDOT:PSS. Although performance losses from increased self-p-doping and non-radiative recombination occur on slower timescales than mobile ion-redistribution-related losses, they too must be addressed to achieve stability commensurate with real-world operation for decades. Slowing the formation of the δ-CsSnI₃ phase, which we observed after weeks to months of aging, is of less urgency but should ultimately also be suppressed.

The emerging picture is that the lead-tin perovskite phase is quite stable under combined heat and light stressing when well-protected from air exposure. However, small-scale changes in defect chemistry strongly affect device performance by causing an increase in non-radiative recombination and self-p-doping, as well as an increased density of mobile iodine vacancies. If acidic PEDOT:PSS can be avoided and self-p-doping further slowed down, we expect that lead-tin PSCs will be no less stable than standard lead-based PSCs. These results are highly encouraging for the realization of efficient and stable multi-junction thin-film PSCs and define a clear direction for future efforts to improve the stability of lead-tin PSCs.

## Methods
### Preparation of perovskite precursor
To make a 1.8 M FA₀.₈₃Cs₀.₁₇Pb₀.₅Sn₀.₅I₃ perovskite precursor, FAI (1.49 mmol, Greatcellsolar), CsI (0.31 mmol, Alfa Aesar, 99.9%), PbI₂ (0.90 mmol, Thermo Fisher Scientific, ultra dry 99.999%), SnI₂ (0.90 mmol, Thermo Fisher Scientific, ultra dry 99.999%), SnF₂ (0.09 mmol, Aldrich, 99%), and metallic Sn powder (10 mg, Aldrich, 99.5%) were stirred in DMF (0.800 ml, Sigma Aldrich, anhydrous) and DMSO (0.200 ml, Sigma Aldrich, anhydrous) for 4 days at room temperature in a glovebox. Solutions were filtered with a 0.45 µm PTFE filter shortly before spin coating.

### Fabrication of films and devices
Glass or ITO substrates (Biotain, 10–15 Ω cm⁻²) were cleaned by scrubbing with dishwashing soap, then sonicated for 10 min in 1 vol% Decon90 in DI water. Substrates were then rinsed with DI water and sonicated in DI water for 10 min, then sonicated in acetone and iso-propyl alcohol (IPA) for 5 min each. Substrates were dried with N₂ and exposed to UV ozone for 10 min immediately before further processing.

For films deposited on glass or PTAA, a layer of Al₂O₃ nano-particles was added to the substrate to improve wetting. Al₂O₃ nano-particle suspension (90 µl, Sigma Aldrich, <50 nm, 20 wt% in IPA, diluted 1:150 in IPA) was deposited on the substrate by dynamic dropping during spinning at 5000 rpm for 30 s and subsequently dried for 2 min at 100 °C.

PEDOT:PSS (Heraeus, AL3083) was mixed with IPA in a 1:2 ratio and filtered with a 0.40 µm PVDF filter shortly before spin-coating. PEDOT:PSS solution (250 µl) was statically dropped and spread onto the ITO substrate, then spin-coated at 4000 rpm and 1333 acc for 30 s. Films were then annealed at 150 °C for 15 min in air, and a further 15 min in a nitrogen-filled glovebox. PTAA (Xi'an Polymer Light Corp, Mw 10,000–100,000 g mol⁻¹) in toluene (1.5 mg ml⁻¹) was stirred overnight to dissolve, then filtered with a 0.45 µm PTFE filter. PTAA was spin-coated onto ITO substrates in a nitrogen glovebox by dynamically dripping 150 µl solution onto the substrate at the start of a 6000 rpm, 1000 acc, 30 s spin-coating cycle. Films were then annealed at 100 °C for 10 min. To improve wetting on the PTAA surface, once the substrates had cooled Al₂O₃ nanoparticles were deposited as described above.

To deposit perovskite films, perovskite precursor (80 µl) was statically dropped and spread onto the substrate, then spun at 5000 rpm and 1000 acc for 60 s. Anisole (200 µl, anhydrous, Sigma Aldrich) antisolvent was dropped onto the substrate after 30 s. The substrates were then annealed at 100 °C for 15 min.

For full device fabrication, the perovskite film was passivated with EDAI₂. EDAI (0.5 mg ml⁻¹, Merck) was stirred in 1:1 toluene:IPA for 4 days and subsequently filtered with a 0.45 µm PTFE filter. 80 µl of this solution was dropped onto the perovskite surface, and once the

solution had spread to the edges of the film, it was spun at 5000 rpm and 5000 acc for 20 s and annealed at 100 °C for 1 min.

PCBM (20 mg ml$^{-1}$, Ossila) was dissolved in 1:3 chlorobenzene:dichlorobenzene by stirring overnight, and filtered with a 0.45 μm PTFE filter. PCBM solution (80 μl) was dynamically deposited on the perovskite whilst spinning at 2000 rpm for 20 s and subsequently annealed at 100 °C for 3 min. BCP (0.5 mg ml$^{-1}$, Xi'an Polymer Light Technology Corp) was dissolved in IPA by stirring for 4 days and filtered with a 0.45 μm PTFE filter. BCP solution (80 μl) was dynamically deposited on the perovskite whilst spinning at 5000 rpm for 30 s and subsequently annealed at 100 °C for 1 min.

Finally, 3.5 nm Cr and 100 nm Au were sequentially thermally deposited onto the device (initial rate 0.02 nm s$^{-1}$, ~10$^{-7}$ Pa vacuum). Films and devices were not exposed to air at any point during the fabrication process.

### Aging

Samples were encapsulated with a glass slide attached to the substrate with UV-activated epoxy, which was cured for 3 min (Everlight Eversolar AB-341). For films, a recessed cavity glass with epoxy only deposited at the encapsulation edge was used to allow for optical measurements, and for devices, the active area was fully covered by the epoxy. Before encapsulation, perovskite material was removed at the epoxy edge for optimal adhesion. Samples were aged in an ambient, illuminated aging chamber (Atlas Suntest XLS+) at 65 °C and simulated full-spectrum sunlight (76 mW cm$^{-2}$) irradiance under open-circuit conditions. Before any measurement was performed, samples were removed from the aging environment and allowed to come to room temperature without direct illumination for 20 min. For XRD and SEM measurements of aged samples, the glass edges of the sample were cut to remove the encapsulation glass and epoxy.

### J−V measurements

The J−V characteristics of devices were measured under AM1.5G illumination (WaveLabs Sinus−220 solar simulator) with 100 mW cm$^{-2}$ equivalent irradiance (certified by KG3-filtered Si reference photodiode). Voltage was swept from 0.9 V to −0.2 V and back at a rate of 0.61 V s$^{-1}$. The device areas were defined by shadow masks, and each substrate contained 3 devices with an area of 0.25 cm$^2$ and one device with an area of 1.00 cm$^2$. For steady-state $V_{oc}$ and $J_{sc}$ measurements, devices were held at 0 V or $V_{oc}$ for 30 s. For steady-state PCE measurements, an MPP tracker based on a gradient descent algorithm was employed to measure the maximum power for 60 s. A quasi-steady-state FF was calculated by dividing the product of V and J at the max power measured by the steady-state PCE measurement by the product of steady-state $V_{oc}$ and $J_{sc}$.

### UV-vis absorption measurements

The total transmittance and total reflectance of encapsulated samples were measured in a Cary 5000 spectrophotometer using an internal diffuse reflectance accessory. Absorbance was calculated according to Eq. 1.

$$A = - \ln(1 - T - R) \qquad (1)$$

### Optical microscope imaging

Samples were imaged in an optical microscope (Nikon Eclipse LV100ND) using a 20× objective in bright-field mode. Samples were illuminated from the bottom.

### Scanning electron microscope and energy dispersive X-ray spectroscopy measurements

SEM images were obtained with a FEI Quanta 600 FEG SEM using an accelerating voltage of 2, 5 or 20 kV as specified in the main text.

Images were always collected on fresh areas to avoid degradation under the electron beam. EDX-SEM maps were acquired with the same FEI Quanta 600 FEG SEM, operating at an accelerating voltage of 20 kV at a working distance of 10 mm. The total acquisition time was 7 min to minimize beam damage.

Note on contrast: We observed the strongest contrast between the bulk and the new crystallites observed at a low imaging voltage of 2 keV. Significant Z-contrast between $FA_{0.83}Cs_{0.17}Pb_{0.5}Sn_{0.5}I_3$ and δ-$CsSnI_3$ is not expected at normal accelerating voltages of 5–20 keV due to similar electron backscattering coefficients (η = 0.433 and 0.423, respectively, calculated as in Goldstein et al. [70]). The stronger contrast between the bulk and the new crystallites observed at the lower voltage of 2 keV may hence be due to topography or local compositional differences on the surface[70]. The darker appearance of the perovskite around these grains in SEM images under low accelerating voltages could suggest a difference in surface composition or roughness.

### X-ray diffraction measurements

XRD measurements were performed on a Panalytical X'pert Pro XRD diffractometer using a Cu-k(alpha) radiation source with a wavelength of 1.54 eV and a generator voltage of 40 V and current of 40 mA. To track any effects related to oxidation during measurements, we perform several successive scans on each sample. During the 2.5-h measurement duration we did not observe formation of additional peaks, but did observe a very slight (<0.1°) peak shift of the perovskite phase to lower angles, which can be explained by the formation of a slightly more FA- and Pb-rich primary phase as small amounts of $Cs_2SnI_6$ or δ-$CsSnI_3$ evolve from the initial perovskite composition.

### Time-resolved photoconductivity measurements

Films were removed from aging conditions 20 min before each measurement and allowed to cool to room temperature in dark conditions. We used the method developed in recent work by Lim et al. [39] A Nd:YAG laser (Ekspla NT342A) excitation source tuned to a wavelength of 600 nm and pumped at 10 Hz with 3.74 ns pulses (full-width-half-maximum, FWHM) was used, attenuated by an optical density filter to achieve a range of fluences. This pulsed light illuminated the entire sample area of 0.25 cm$^2$ to uniformly excite the film from the substrate side. A photodetector (Thorlabs, FDS015) was used to detect the pulse source for the optical trigger for transient measurements using a digital oscilloscope. A small DC bias (<5 mV μm$^{-1}$, which is three orders of magnitude smaller than that for solar cell characterization, ~3 V μm$^{-1}$) was applied across the in-plane (lateral) electrodes (300 or 500 μm lateral spacing), while the current was monitored by an oscilloscope. As the contact resistance between the perovskite film and Au electrode is fairly small compared to the sample resistance, we employed a two-wire conductivity measurement. A fixed resistor was put in series with the sample in the circuit to be <1% of the sample resistance. We monitored the voltage drop across the variable series resistor through a parallel oscilloscope (1 MΩ input impedance) to determine the potential drop across the two in-plane Au electrodes on the sample. The measured photoconductivity value was obtained after 1 min illumination to minimize the time-dependent photo-doping (total measurement time for one sample is <5 min). Transient conductivity (σ) was calculated by Eq. 2, where $V_R$ is the monitored voltage drop through the fixed resistor, $R_R$, $V_{app}$ is the applied bias voltage, l is the channel-to-channel length, w is the channel width, and t is the film thickness. The dark conductivity was determined by taking the mean conductivity measured during the 320 ns before the light pulse.

$$\sigma = \frac{V_R}{R_R(V_{app} - V_R)} \bullet \frac{l}{wt} \qquad (2)$$

Significant recombination of photogenerated charges is expected to already take place during the length of the light pulse (3.74 ns). We

partially corrected for this by fitting the photoconductivity decay to a mono-exponential decay and extrapolating this back to the beginning of the pulse ($t_0$) to determine the maximum photoconductivity reached. However, higher-order recombination as well as diffusion- and trap filling-related processes are also expected to take place during the pulse duration, which we do not correct for here. Hence, the photoconductivity and hence the sum of mobilities that we determine are likely underestimates and should be understood as lower bounds. We measured the photoconductivity at multiple fluences and selected the fluence that results in the highest sum of mobilities calculated.

The extraction of the sum of free electron and hole mobilities and background hole density is detailed below. The conductivity ($\sigma$) measured can be expressed as

$$\sigma = e(\mu_n(\Delta n + n_0) + \mu_p(\Delta p + p_0)) \tag{3}$$

where $n$ and $p$ are the electron and hole densities, and $\mu_n, \mu_p$ the electron and hole mobilities, and $e$ is the elementary charge. As lead-tin perovskites are expected to be p-type, $n_O$ will be many orders of magnitude smaller than the photogenerated charge carrier density $\Delta n$ (here in the range of $10^{16}$–$10^{19}\,\text{cm}^{-3}$), allowing us to approximate $\Delta n + n_O \approx \Delta n$[71]. From the maximum photoconductivity $\sigma_{photo}$, dark conductivity, and photogenerated charge carrier density $\Delta n$ at $t_O$, the sum of charge carrier mobilities is estimated according to Eq. 4.

$$\frac{\sigma_{photo} - \sigma_{dark}}{e\Delta n} = \mu_n + \mu_p \tag{4}$$

From the sum of mobilities and the measured dark conductivity, we can estimate an upper limit for $p_0$ as in Eq. 5.

$$\frac{\sigma_{dark}}{e\mu_p} = p_0 \tag{5}$$

**Photoluminescence quantum efficiency measurements**
Films were removed from aging conditions 20 min before each measurement and allowed to cool to room temperature in dark conditions. PLQE of samples was determined according to the method of de Mello et al.[72] Samples were placed inside an integrating sphere and excited from the substrate side with a 657 nm continuous wave laser excitation source (Thorlabs) with a large spot size of 0.15 cm². The resulting PL signal was collected via a fiber bundle (Ocean Optics QR600 7 SR125BX) coupled with a spectrometer (QE Pro, Ocean Optics), and a stray light correction was applied to the recorded spectra after measuring. Two different spots were measured on each substrate. QFLS was calculated with the following equation[73]:

$$QFLS = QFLS_{rad} + k_B T \ln(\text{PLQE}) \tag{6}$$

**PLQE simulations**
The change in carrier density in space and over time was simulated using the equations below. The processes of 1D diffusion, Auger recombination, bimolecular radiative recombination, and single-electron trap-assisted non-radiative recombination were simulated. Calculation of the various parameters and recombination rates was performed via Eqs. 7–15.

$$\frac{dn}{dt}(t) = G(t) - D_n \frac{d^2 n}{dx^2}(t) - R_{aug}(t) - R_{rad}(t) - R_{trap1,n}(t) \tag{7}$$

$$\frac{dp}{dt}(t) = G(t) - D_p \frac{d^2 p}{dx^2}(t) - R_{aug}(t) - R_{rad}(t) - R_{trap1,p}(t) \tag{8}$$

$$\frac{dn_{trap}}{dt}(t) = R_{trap,n}(t) - R_{trap,p}(t) \tag{9}$$

$$n_0 = 2\left(\frac{m_n kT}{2\pi\hbar^2}\right)^{\frac{3}{2}} e^{\frac{-(E_{CB}-E_F)}{kT}} \quad p_0 = 2\left(\frac{m_p kT}{2\pi\hbar^2}\right)^{\frac{3}{2}} e^{\frac{-(E_F-E_{VB})}{kT}} \tag{10}$$

$$n = n_0 + \Delta n, \quad p = p_0 + \Delta p \tag{11}$$

$$R_{aug} = (k_{aug,n} n + k_{aug,p} p)(np - n_0 p_0) \tag{12}$$

$$R_{rad} = k_{rad,ext}(np - n_0 p_0) \tag{13}$$

$$R_{trap1,n} = \beta_n n(N_t - n_t) - \beta_n \left(N_C e^{\frac{E_T - E_{CB}}{kT}}\right) n_t \tag{14}$$

$$R_{trap1,p} = \beta_p p n_t - \beta_p \left(N_V e^{\frac{E_{VB} - E_T}{kT}}\right)(N_t - n_t) \tag{15}$$

To simulate PLQE, the simultaneous system of Eq. 16 was solved for steady-state carrier densities $n(x)$ and $p(x)$. The trap-assisted non-radiative recombination rate and external PLQE were then calculated as in Eqs. 17, 18.

$$\frac{dn}{dt}(t) = 0 \quad n - n_0 + n_{trapped} = p - p_0 + p_{trapped} \tag{16}$$

$$R_{SRH} = \frac{N_t \beta_n \beta_p (np - n_0 p_0)}{(n + N_C e^{\frac{E_T - E_{CB}}{kT}})\beta_n + (p + N_V e^{\frac{E_{CB} - E_T}{kT}})\beta_p} \tag{17}$$

$$PLQE = \frac{R_{rad}}{R_{aug} + R_{rad} + R_{SRH}} \tag{18}$$

**External quantum efficiency measurements**
The external quantum efficiency of our devices was determined using Fourier transform photocurrent spectroscopy. Our custom-built setup is based on a Bruker Vertex 80 V Fourier transform interferometer. The solar cells were masked with a metal aperture such that the whole active area was illuminated by a tungsten halogen lamp. To determine the EQE, the photocurrent spectrum of the device under test was divided by that of a calibrated Si reference cell (Newport) of a known EQE. The acquisition time for each photocurrent spectrum was ~60 s.

To determine the equivalent short-circuit current density under 1 sun irradiance from the EQE measurements, the overlap integral of the AM1.5 photon flux ($\varphi_{AM1.5}$) spectrum with the EQE was calculated. Explicitly, this is given by

$$J_{sc} = q \int_0^\infty d\lambda\, EQE(\lambda)\varphi_{AM1.5}(\lambda) \tag{19}$$

where $q$ is the elementary charge and $\lambda$ is the wavelength.

Bandgaps were determined from EQE by plotting $(E*EQE)^2$ against E, then finding the point at which a linear fit to the EQE onset goes to 0[74].

## Variable-rate *J*−*V* scanning/fast hysteresis measurements

To investigate the transient impact of mobile ions on our devices, an in-house-built fast JV setup was used.

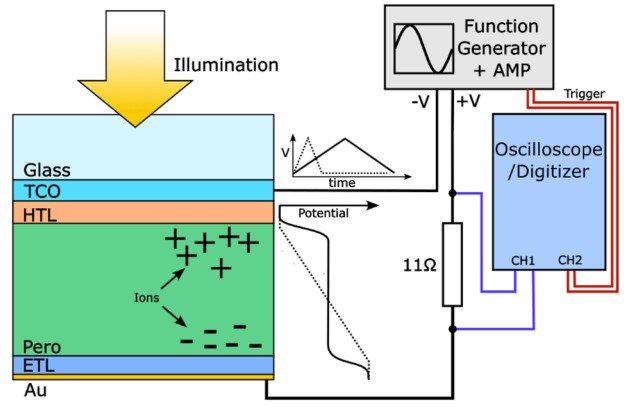

The device was connected to a function generator (RSDG 1032X) coupled to a generic operational amplifier, with a 10 ohm resistor placed in series. The forward and reverse voltage sweep was induced by sending a triangular pulse at a set frequency to the device, with the amplifier providing the power to drive the cell. The current was measured by measuring the potential drop across the resistor in series, which was converted to a current by Ohm's law. This was done using a digitizer (Picoscope 204A), resulting in a current-to-time trace. A trigger pulse from the function generator was coupled into the digitizer, allowing us to link the trace time to the applied voltage to infer the JV curve. JV curves were measured over a large range of frequencies in logarithmically spaced intervals. Devices were pre-biased at 1.1 V for 5 s, before scanning to −0.2 V and back to 1.1 V at the specified rate. Whilst the pre-biasing voltage was selected to lie close to the $V_{oc}$, this may have moved ions into a slightly more or less favorable position for optimal device performance. To reduce noise, 10 repeats for each applied frequency were taken and averaged, and a spline fit to the *J*−*V* curves was performed to extract parameters ($V_{oc}$, $J_{sc}$, FF, PCE).

## *J*−*V* scan simulations

We carried out *J*−*V* curve simulations using both the IonMonger[56] and the SCAPS−1D (v 3310)[65] drift-diffusion packages. The full list of the parameters used (when not varied) is given in Tables S3 and S4. The generation profile used was calculated using a transfer matrix optical model, adjusted to match the short-circuit current measured.

## Reporting summary

Further information on research design is available in the Nature Portfolio Reporting Summary linked to this article.

# Data availability

The data generated and used in this study are available within the paper, its supplementary information files, or deposited in the Oxford Research Archive (https://doi.org/10.5287/ora-vq12opeej).

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

## Acknowledgements

F.M.R. was funded by the EPSRC, UK, under grant number EP/S516119/1. A.D. expresses his gratitude to the Penrose scholarship for very generously funding his studentships. M.K.C. would like to thank the DFG for financial support via the SPP2196 Priority Program (CH 1672/3-1). H.J. acknowledges the support of the sponsorships from Oxford PV Ltd.

## Author contributions

F.M.R. and H.J.S. conceived this project. A.D. carried out fast hysteresis measurements and part of the SCAPS simulations. M.K.C. carried out time-resolved photoconductivity measurements. H.J. fabricated part of the samples. J.A.S. carried out SEM-EDX measurements. H.J., M.F., and F.M.R. designed material and device fabrication methods. F.M.R. carried out all other experiments and data analysis. H.J.S. funded and supervised this project. J.M.B. and J.A.S. provided additional supervision. F.M.R. wrote the paper. All authors contributed to the discussion and reviewed the manuscript.

## Competing interests

H.J.S. is co-founder and chief scientific officer of Oxford PV Ltd., a company commercializing perovskite PV technology. J.M.B. is co-CEO of a company commercializing test and measurement equipment for solar cell research. The remaining authors declare no competing interests.
