## [Transparent Peer Review file · Nature Communications]

Disentangling Degradation Pathways of Narrow Bandgap Lead-Tin Perovskite Material and Photovoltaic Devices

Corresponding Author: Professor Henry Snaith

Version 0:

Reviewer comments:

Reviewer #1

(Remarks to the Author)

In this manuscript, the authors suggested that, under ISOS L-2 aging conditions, the decay of FA_{0.83}Cs_{0.17}Pb_{0.5}Sn_{0.5}I₃ film on PEDOT:PSS is dominated by increased mobile ions, but the decay of FA_{0.83}Cs_{0.17}Pb_{0.5}Sn_{0.5}I₃ film on PTAA is not. Although this work suggests the importance of HTL for the stability of Pb-Sn perovskites, but unfortunately, that's as far as it went. In my opinion, ion migration effect can be increased by varied reasons. This work did not really reveal a specific factor that dominates the decay of Pb-Sn perovskites, or how the decay could be suppressed. This manuscript needs to be substantially improved to match the journal quality of Nature Communications.

The following are my detail comments to this study:

1. Knowing that the decay of FA_{0.83}Cs_{0.17}Pb_{0.5}Sn_{0.5}I₃ film on PEDOT:PSS (or PTAA) is dominated by increased mobile ions (or not) is publishable, but not sufficient for journals such as Nat. Commun. at this stage. The authors did not clarify what mechanism induced the difference in ion migration, and why this difference will be important. Le Corre et al. innovated the fast hysteresis measurement to identify the PCE loss (i.e. the "ionic loss") caused by redistributing mobile ions in aged perovskite solar cells (Sol. RRL 6, 2100772 (2022)). Then, Jarla et al. applied this method and suggested that ionic loss could be common in various aged perovskite solar cells, and meanwhile, it is also sensitive to the used HTL (Nature Energy 9, 664 (2024)). In my opinion, this study added another example (i.e., the FA_{0.83}Cs_{0.17}Pb_{0.5}Sn_{0.5}I₃ film on PEDOT:PSS or PTAA). Although a great effort has been made to compare the aged perovskite films on PEDOT:PSS and PTAA, including the changes in film morphology, the formation of impurity phases, I-V curves at different scanning speeds, the PLQE, doping concentrations, etc. However, no significant new findings or in-depth understanding have been made so far.
2. In the case of perovskite film on PTAA substrate, the authors claimed that "the impact of mobile ions is significantly mitigated". This statement needs stronger supports. Indeed, Figure S10 shows that the device using PTAA HTLs show similar low PCE values at both "slow" scans and "fast" scans conditions. However, this data cannot directly prove that the PCE loss at "fast" scans is not related to mobile ions. Even in Jarla's recent study (Nature Energy 9, 664 (2024)), those mobile ions, which do not respond to very fast scanning, were not excluded as a source of PCE loss. One possible situation is that, some mobile ions may have travelled into charge transport layers during aging condition and it don't return back to perovskite film at pre-bias condition. (This possible situation is also consistent with the increased doping level in the aged perovskite films on PTAA HTL, BTW).
3. This study focused on the impacts of HTLs on the stability of Pb-Sn perovskites. However, only two HTLs were studied. The scope of this study is narrow, which may be the reason why no clear scientific assumption or general conclusion can be drawn from the current results.
4. In page 4, a major statement of this study "We additionally reveal that the magnitude of this mobile ion-induced loss can be greatly reduced by selecting an alternative HTL" is misleading and questionable. First, as mentioned above, the authors did not directly prove the PCE loss in PTAA case is not related to mobile ions; second, the device based on PTAA HTLs still suffers from a serious PCE loss after ageing, the origin of which is unfortunately still unknown, i.e., PTAA HTLs did not provide better stability than that of PEDOT:PSS HTLs. At this situation, it's inappropriate to give the impression that perovskite on PTAA HTLs is stable. Same misleading statement was made in page 20, lines 503-505.
5. Generally, substrates could play important role on the crystallinity or vertical phase separation of perovskites films. It's necessary to explain how the possible influence of crystallinity or vertical phase separation on the film stability (e.g., see a recent paper DOI: 10.1002/adfm.202402655) and PLQE can be excluded? For example, the QFLS (obtained from the PLQE of perovskite on glass) is higher than the Voc (obtained from the device based on PEDOT:PSS and PTAA) in Figure 3b, which was explained as "interfacial losses". However, the non-radiative recombination centers in the perovskite films bulk

can also be much influenced by the variations in crystallinity caused by different substrates. A similar concern can be raised about the different decay channels observed in the device based on PEDOT:PSS and PTAA.

6. How the photogenerated charge carrier densities (in equation 3) were measured? And how to prove that the mobility in dark condition (with traps incompletely filled) is close to that in the light condition (with traps filled)? Please explain it.
7. In page 12, the statement of “bulk recombination dominates over interfacial recombination at the PTAA/perovskite interface” is not supported by experimental results. Besides, the statement of “a significant reduction in the gradient of the PLQE against excitation intensity of PTAA/perovskite stacks (Figure 2c), which could be due to a significant increase in the background carrier density of the perovskite.” is not supported by experimental results neither. This statement is unreasonable because the PLQE of PTAA/perovskite sample is still stronger than that of glass/perovskite sample after 288 h ageing, which cannot be explained by the increased p-type doping.
8. In page 15, the authors mentioned that “Devices using PTAA are more stable during the first 100 hours of aging”. This description is inappropriate because both kinds of devices (using PTAA or PEDOT:PSS) show significant decay, and the differences in the normalized PCE at 100 hours are more likely in the range of error.
9. There are some ambiguous descriptions in page 20, between line 493-501. The discussion is difficult to understand, and poorly supported by experimental results.

The input of this study is more incremental rather than disruptive. I don't believe it fits with the standards of Nat. Commun. in current version.

Reviewer #2

(Remarks to the Author)

The manuscript explores the causes of instability of tin-lead perovskite solar cells under combined heat and light stress (ISOS L-2 conditions). It aims to address the questions by disentangling the degradation occurring in various components in the device stack and in fully fabricated solar cells. Findings show that lead-tin perovskite films are stable beyond the usual timescales associated with device degradation. Mobile ions are the major cause of early-time performance degradation, and it can be mitigated by selecting an alternative hole transport layer. However, the authors do not provide strong and sufficient evidence for some findings and statements. Overall, the work needs further major revision before consideration for publication in Nature Communications. Some detailed comments are as follows:

1. To illustrate that the new phase peaks are ascribed to the appearance of degradation products, the XRD patterns of fresh ITO/PEDOT: PSS or ITO/PTAA/perovskite samples should be provided to present their differences with aging ones (aged 2160 hrs encap, 1 sun, 65°C. Figure S2).
2. For cubic perovskite cells fitted to PbSn-perovskite films with same stacks of Glass/Al₂O₃ NPs/FA_{0.83}Cs_{0.17}Pb_{0.5}Sn_{0.5}I₃, even though they show respective slight expansion of cubic lattice volume after aging under conditions of “unencapsulated ambient, room temperature” and “unencapsulated in N₂, 65°C, 0.76 suns”, the same stacks of fresh samples should have same unit cell volumes, but they show different parameters in Table S1. Please give an explanation.
3. In 184-186 lines, authors state that “we observe voids filled with smaller crystallites in the active layer of aged devices (Figure 1d), which we don't observe in fresh devices”, but no direct evidences (e.g., cross-section SEM image of fresh devices, etc) are not presented in the manuscript and supplementary file. Please provide the corresponding results to evidence the claim.
4. Some descriptions in the manuscript are not well evidenced. For example, the content in 212-213 lines and the responding conclusion state that “although the perovskite's bulk structure and optical absorption properties remain stable during the first few hundred hours of aging”. From the provided OM and SEM results, we can know that the perovskite in the device changed after 625 h aging. However, the results of OM and SEM proves that perovskite's bulk structure remaining stable during the first few hundred hours of aging are absent in the manuscript. What is the longest time that the perovskite structure maintains no change? The authors shall provide the corresponding evidences.
5. In 322-323 lines of the manuscript, the authors state “we find that the p₀ and mobility of isolated lead-tin perovskite films on glass are stable over 300 hours of aging”. However, the authors only show the background hole density and mobility for the glass/perovskite samples upto 150 hours (Figure 2). More comprehensive results shall be provided to support their claims.
6. In Figure 2c, the power intensity-dependence of glass/perovskite sample aging 288 h is missing. Meanwhile, the PLQE of PEDOT: PSS/perovskite films by aging is always higher than fresh. The authors should provide comprehensive results and detailed reasons for this unusual phenomenon.
7. The dark conductance of glass/PEDOT: PSS/perovskite stack does not show significant changes after 20 h (Figure 2d), but that of glass/PTAA/perovskite stack shows a significant increase of 50 fold in the aging period from 100 hours to 300 hours. Authors infer that the presence of PTAA during aging induces increased p-doping of lead-tin perovskite. How about the sample Glass/perovskite between 100-300 hours of aging?
8. In this work, the authors reveal that the early-stage device degradation of lead-tin PSCs is dominated by mobile ions on the device under heat and light stressing, which can be mitigated using PTAA instead of PEDOT:PSS. However, devices

using PTAA experience much larger non-mobile-ion related FF and Jsc losses, which result in worse steady-state PCE after 288 hours of aging. Therefore, according to contents in current manuscript, it is still not good to use PTAA replacing PEDOT:PSS, and it is still not clear that which hole transport layer is a good replacing alternative of PEDOT:PSS, even though the underlying interface problems are pointed. Does authors try and compare other popular HTL materials, such as NiOx, SAM, to more clearly reveal the degradation problems in tin-lead perovskite devices and offer the good choice of alternative HTL for degradation-mitigation?

Reviewer #3

(Remarks to the Author)

In this work, Snaith et al studied the degradation behavior of lead-tin based perovskite solar cells, and claim that the oxidation of Sn²⁺ to Sn⁴⁺ is not a big issue. Instead, the ion migration /accumulation is the major reason underlying the device degradation, but such effect can be mitigated by using different charge transporting layers. The authors present comprehensive and persuasive evidences to strongly support their conclusions. While, much better stability of tin-lead mixed perovskite compared to the pristine tin-based counterparts has been widely observed and reported. Therefore, it will be critically important if this work can present some insightful understandings on the origins at atomic levels. Therefore, I would like to consider publication of this work after addressing the following issues.

- 1) The pure tin and lead based perovskite films and devices should be also studied to compare the results with the lead/tin mixed perovskite. This will be helpful to understand the working mechanisms of the improved stability.
- 2) A further step toward understanding the working mechanism is suggested to be presented, e.g. unveiling it at an atomic level.
- 3) Vertical carrier mobility in a perovskite solar cells is more important compared to the lateral carrier mobility. However, this work only presents the lateral mobility. Therefore, the carrier mobility along the vertical direction should be examined.
- 4) Buried interface morphology should be studied to correlate the degradation of perovskite film and device.
- 5) The authors claim the ion migration and accumulation is the key governing the degradation of lead-tin based devices. However, this work lacks some essential evidences to support this point. The forward and reverse J-V scans at different speeds is not persuasive enough.

Version 1:

Reviewer comments:

Reviewer #1

(Remarks to the Author)

In this revised manuscript, the decay processes of Pb-Sn perovskites on PEDOT:PSS and PTAA have been analysis and described more appropriately. After modification, the contribution of this study has been better clarified. What convinces and attracts me most is their reformulated statement of: the chemical interaction between the perovskite and PEDOT:PSS counteracts the detrimental p-doping effect that was observed in PTAA/Perovskite case. Since this idea, the rest of the discussion become innovative and reasonable. The authors have collected a lot data to compare the different decay process between "PEDOT:PSS /Perovskite" and "PTAA/Perovskite", but lacks a clear idea in their first version of manuscript. By admitting that "although lead-tin PSCs employing PTAA are less affected by mobile ion redistribution-related losses during aging, an increase in p-doping during aging leads to even worse overall degradation", in my opinion, the authors are accessing a more attractive conclusion, i.e., the oxidation of Sn in the "PTAA/Perovskite" case is more obvious than that in the case of "PEDOT:PSS /Perovskite", and this oxidation can be weakened by the presence of iodine vacancies induced by PEDOT:PSS. So, at this point, it is reasonable to speculate that the oxidation of Sn might be related to the dynamically generated, reactive iodine.

This revised manuscript can be accepted, but it is lengthy and difficult to read. I suggest the authors try their best to reorganize the writing.

Reviewer #2

(Remarks to the Author)

This work studies the degradation occurring in various components in the device stack and fully fabricated solar cells. Findings show that lead-tin perovskite films are stable beyond the usual timescales associated with device degradation. Mobile ions are the major cause of early-time performance degradation, and it can be mitigated by selecting an alternative hole transport layer. The authors have provided evidence to support the findings and statements. The work can be considered for publication in Nature Communications after addressing the following issues.

1. For the statement "we observe voids filled with smaller crystallites in the active layer of aged devices (Figure 1d), which we don't observe in fresh devices", the authors have added the evidence of cross-section SEM image of fresh devices in Figure S3. It is suggested to merge Figure S3 into Figure 1d to show the comparison of fresh and aged C-SEM images.

2. The authors answered that XRD and SEM measurements were not performed at intermediate timesteps due to the difficulty of removal of encapsulation and destruction of the device. However, if the exact timeline of perovskite structure change is uncertain, the authors should revise the expression according to their experimental results. The authors shall add

wording (e.g., it is a hypothesis that ...) in the statements without experimental proofs.

3. In Figure 2a, the annotations of blue and purple lines are the same ("glass/perovskite"). Please check and correct them.

4. There are some format errors in the text, like "288 hours", and reference format errors. Please carefully check and correct.

Reviewer #3

(Remarks to the Author)

The paper was well revised, and I suggest the acceptance of it for publication.

REVIEWER COMMENTS

Reviewer #1 (Remarks to the Author):

In this manuscript, the authors suggested that, under ISOS L-2 aging conditions, the decay of FA_{0.83}Cs_{0.17}Pb_{0.5}Sn_{0.5}I₃ film on PEDOT:PSS is dominated by increased mobile ions, but the decay of FA_{0.83}Cs_{0.17}Pb_{0.5}Sn_{0.5}I₃ film on PTAA is not. Although this work suggests the importance of HTL for the stability of Pb-Sn perovskites, but unfortunately, that's as far as it went. In my opinion, ion migration effect can be increased by varied reasons. This work did not really reveal a specific factor that dominates the decay of Pb-Sn perovskites, or how the decay could be suppressed. This manuscript needs to be substantially improved to match the journal quality of Nature Communications.

We thank the reviewer for raising that the impact of the aggregation of mobile ions at interfaces can be increased by multiple factors. We have now added extensive simulations to investigate the effect of a decrease in bulk lifetimes, increase in surface recombination velocity, and change in mobile ion density on J-V parameters at a range of scan rates, and compared these to our experimental data. These additional experiments have strengthened the work and enabled the uncovering of a mechanism by which PEDOT:PSS and PTAA affect device performance differently during aging. The additions to the manuscript are further detailed in our responses below.

The following are my detail comments to this study:

1. Knowing that the decay of FA_{0.83}Cs_{0.17}Pb_{0.5}Sn_{0.5}I₃ film on PEDOT:PSS (or PTAA) is dominated by increased mobile ions (or not) is publishable, but not sufficient for journals such as Nat. Commun. at this stage. The authors did not clarify what mechanism induced the difference in ion migration, and why this difference will be important. Le Corre et al. innovated the fast hysteresis measurement to identify the PCE loss (i.e. the “ionic loss”) caused by redistributing mobile ions in aged perovskite solar cells (Sol. RRL 6, 2100772 (2022)). Then, Jarla et al. applied this method and suggested that ionic loss could be common in various aged perovskite solar cells, and meanwhile, it is also sensitive to the used HTL (Nature Energy 9, 664 (2024)). In my opinion, this study added another example (i.e., the FA_{0.83}Cs_{0.17}Pb_{0.5}Sn_{0.5}I₃ film on PEDOT:PSS or PTAA). Although a great effort has been

made to compare the aged perovskite films on PEDOT:PSS and PTAA, including the changes in film morphology, the formation of impurity phases, I-V curves at different scanning speeds, the PLQE, doping concentrations, etc. However, no significant new findings or in-depth understanding have been made so far.

We thank the reviewer for their comments and the recognition of our efforts to investigate all possible modes contributing to the degradation of lead-tin perovskite solar cells. We agree with the reviewer that deeper investigation of the mechanism inducing the different behaviours observed for the different HTLs would greatly improve the impact of the work.

To this end, we have now performed extensive variable-rate J-V scan simulations using the IonMonger drift-diffusion package (figures 5 and S15). These simulations reveal that the behaviour observed in devices using PTAA across the full range of scan rates can be explained by a decrease in bulk lifetime, whilst that of devices using PEDOT:PSS can only be replicated by including an increase in rapid-moving ion density. These observations are explained by previous findings that acidic PEDOT:PSS reacts with iodine in the perovskite to form HI, leaving behind highly mobile iodine vacancies.¹

This mechanism is also consistent with the increased p-doping observed in devices using PTAA HTLs. The generation of positively charged iodine vacancies (donor defects) in devices using PEDOT:PSS counteracts potential self-p-doping in the lead-tin perovskite. We propose that some p-doping process (such as Sn oxidation) occurs in these devices independent of HTL, but is counteracted in devices using PEDOT:PSS by a higher density of iodine vacancies. Hence, all of our observations can be explained by the tendency of PEDOT:PSS to rapidly generate iodine vacancies in the perovskite during aging, a novel and relevant finding for the community. Sections 2.4 and 2.5 have been significantly updated to convey these new findings.

In addition, the relatively slow timescale (2-3 weeks) of self-p-doping we observed in our well-encapsulated lead-tin perovskite films under ISOS-2 aging conditions, as well as the repression of p-doping when PEDOT:PSS is used, have not been revealed before. The abstract, introduction and conclusion of the paper were edited to better highlight the findings made.

2. In the case of perovskite film on PTAA substrate, the authors claimed that “the impact of mobile ions is significantly mitigated”. This statement needs stronger supports. Indeed, Figure S10 shows that the device using PTAA HTLs show similar low PCE values at both “slow” scans and “fast” scans conditions. However, this data cannot directly prove that the PCE loss at “fast” scans is not related to mobile ions. Even in Jarla’s recent study (Nature Energy 9, 664 (2024)), those mobile ions, which do not respond to very fast scanning, were not excluded as a source of PCE loss.

We entirely agree with the reviewer, and have edited the text in section 2.4 to further emphasize that the variable rate J-V measurements only capture the bias-induced effects that occur on timescales between the fast and slow scan rates. As the reviewer suggests, ionic species with low mobilities can still affect device performance by mechanisms other than rapid bias-induced redistribution. We have now better clarified this in the text by referring to the changes in J-V parameters with scan rate as being caused by the redistribution of ‘rapid-moving’ ions.

Furthermore, we have now emphasized that charged point defects/mobile ions don’t disappear at fast scans but are immobile at pre-biasing positions, and that the difference between ‘fast’ and ‘slow’ scans represents the difference between the impact of these ions at their pre-biasing position compared to their redistributed position. We furthermore added analysis of J-V scans collected at a range of scan rates, rather than just considering one fast and one slow scan. Most notably, the text on pg. 16 has been edited to read as follows:

“To investigate how such processes affect device performance during aging, we perform J-V scans at different scan rates on fresh and aged devices.² Devices are pre-biased close to V_{oc} before carrying out a reverse and forward J-V sweep. We reliably measured J-V scans with rates between 0.2 ms - 7s per scan (Figure S13). When a scan is performed at a slower rate than the characteristic time of ion motion, mobile ions are expected to redistribute during the scan in response to the applied voltage. When a scan is performed at a faster rate than the characteristic time of ion motion, however, it reveals the characteristics of a device with the ions immobile in their pre-biasing position.³ The main bias-driven process expected to affect device performance on the ms-s timescale is the migration of iodine vacancies across the perovskite.⁴⁻⁸ Other mobile ion candidates such as A- or B- cation vacancies may still affect

device stability by migrating during aging, but are not expected to undergo bias-driven migration on the timescale of our J-V scans.”

One possible situation is that, some mobile ions may have travelled into charge transport layers during aging condition and it don't return back to perovskite film at pre-bias condition. (This possible situation is also consistent with the increased doping level in the aged perovskite films on PTAA HTL, BTW).

To cause p-doping of the perovskite, positively charged ions such as FA^+/Cs^+ or $\text{Sn}^{2+}/\text{Pb}^{2+}$ must have migrated into the PTAA under open circuit aging conditions. Whilst such a mechanism cannot be excluded, it is unclear what would be the driving force for such migration. We instead propose an alternative explanation for the increased p-doping on PTAA, as explained above and in section 2.5 of the manuscript.

3. This study focused on the impacts of HTLs on the stability of Pb-Sn perovskites. However, only two HTLs were studied. The scope of this study is narrow, which may be the reason why no clear scientific assumption or general conclusion can be drawn from the current results.

We studied the two HTLs that are most commonly used in lead-tin perovskite solar cells. Unfortunately, we have not been able to produce lead-tin solar cells with other commonly used HTLs (such as SAMS) with a comparable PCE to those fabricated on PEDOT:PSS and PTAA. The perovskite quality and the movement of charge carriers through these lower PCE devices is expected to be quite different, which would make it difficult to draw conclusions from their study.

We prioritized elucidating the degradation mechanism of devices using PEDOT:PSS, because it is by far the most commonly used HTL for high-performance lead-tin perovskite solar cells. To do this, comparison with a chemically inert polymeric HTL such as PTAA was sufficient.

4. In page 4, a major statement of this study “We additionally reveal that the magnitude of this mobile ion-induced loss can be greatly reduced by selecting an alternative HTL” is misleading and questionable. First, as mentioned above, the authors did not directly prove the PCE loss in PTAA case is not related to mobile ions; second, the device based on PTAA

HTLs still suffers from a serious PCE loss after ageing, the origin of which is unfortunately still unknown, i.e., PTAA HTLs did not provide better stability than that of PEDOT:PSS HTLs. At this situation, it's inappropriate to give the impression that perovskite on PTAA HTLs is stable. Same misleading statement was made in page 20, lines 503-505.

We did not intend to give the impression that PTAA HTLs provide better overall stability in these devices. We have adjusted the phrasing in the text to clarify that this statement refers to the losses from “*the bias-induced redistribution of rapid-moving mobile ions*” (pg. 17), which occurs within ~1s during a J-V scan. We have also changed our labelling of “*non-ionic losses*” to “*other losses*” (Figure 4b), which encompasses a wide range of processes that are “*factors other than the redistribution of rapid-moving ions*” (pg. 17). The entirety of section 2.4 has been significantly edited to improve this wording.

5. Generally, substrates could play important role on the crystallinity or vertical phase separation of perovskites films. It's necessary to explain how the possible influence of crystallinity or vertical phase separation on the film stability (e.g., see a recent paper DOI: 10.1002/adfm.202402655) and PLQE can be excluded? For example, the QFLS (obtained from the PLQE of perovskite on glass) is higher than the Voc (obtained from the device based on PEDOT:PSS and PTAA) in Figure 3b, which was explained as “interfacial losses”. However, the non-radiative recombination centers in the perovskite films bulk can also be much influenced by the variations in crystallinity caused by different substrates. A similar concern can be raised about the different decay channels observed in the device based on PEDOT:PSS and PTAA.

We thank the reviewer for these useful suggestions and agree that differences in perovskite crystallinity or the vertical phase inhomogeneity mentioned could affect the interpretation of PLQE measurements. There are several reasons we believe this plays a more minimal role in the case of our samples:

1. Efforts were made to ensure substrate surface qualities were as similar as possible, e.g. the PTAA used is expected to be sufficiently thick to smooth the ITO surface, and Al₂O₃ NPs were used as an interlayer when perovskite was deposited on glass and PTAA to improve wetting.
2. We do not observe a difference in the fresh PL peak position (now included in Figure S6), nor any significant differences in XRD or SEM measurements of fresh

glass/perovskite, ITO/PEDOT:PSS perovskite, and ITO/PTAA/perovskite samples (which have now been added to Figure S2).

However, there remains a possible difference in bulk defect density and homogeneity, which can affect PLQE as suggested by the reviewer. We have now included this in our discussion in section 2.2, referencing the paper suggested by the reviewer.

“For the samples deposited on HTLs, the crystallinity (from XRD, Figure S2) and PL peak position (Figure S6) of films were similar, although small differences in film quality or heterogeneity⁹ stemming from the difference in substrate could still contribute to differences in defect formation rates during aging.”

6. How the photogenerated charge carrier densities (in equation 3) were measured?

The photogenerated charge carrier densities were estimated to be equal to the photon flux delivered by the laser pulse (from $t=0-4$ ns). We make an estimate of the conductivity at t_0 by extrapolating a monoexponential fit of the photoconductivity back to t_0 , but expect this to be an underestimate. We have expanded our explanation of this in Supplementary Note 1 in the SI. In this note we also discuss a potential increase in mobility with illumination (SI pg. 9).

“Supplementary Note 1: Dependence of photoconductivity of excitation fluence

To calculate the sum of mobilities from the measured TPC decays, we need to estimate the value of the conductivity immediately after the laser pulse. The maximum conductivity measured (which normally occurs 20-40 ns later) is a significant underestimate of this, as during this time free charge carriers have already undergone significant recombination and have also been removed from the perovskite by charging of the electrodes.¹⁰ In this paper, we partially correct this by extrapolating a stretched mono-exponential fit of the photoconductivity back to t_0 . However, this only corrects for the charge carriers removed by non-radiative recombination.

To probe the effect of this, we measured TPC decays at 9 different fluences and plotted the extracted sums of mobilities in Figure S7. In all samples we see a decrease in the sum of mobilities with higher excitation fluence, as previously observed by other.¹¹ This is consistent

with a stronger underestimation of the t_0 conductivity at higher excitation fluences, as higher order recombination mechanisms become more dominant. We hence select the maximum sum of mobilities calculated over all excitation fluences as the ‘real’ mobility.

Interestingly, the dark conductivity also shown significant fluence dependence in some of our samples. Unlike the sum of mobilities extracted from the photoconductivity, dark conductivity increases with fluence. This could be caused either by the delayed release of charge carriers after excitation (for example by de-trapping from long-lived trap states or capacitive effects¹²) and/or by a long-lived increase in mobility as a result of photoexcitation. To minimize the impact of this effect, we use the dark conductivity measured with the lowest excitation fluence to determine p_0 .

Figure S7 Dark conductivity and sum of mobilities extracted from TPC traces at a range of excitation densities, for a) a glass/perovskite sample (referred to as 'batch 1' in the main text), b) a glass/PEDOT:PSS/perovskite sample, and c) a glass/PTAA/Al₂O₃ NPs/perovskite sample."

And how to prove that the mobility in dark condition (with traps incompletely filled) is close to that in the light condition (with traps filled)? Please explain it.

It is indeed possible that the hole and/or electron mobility changes depending on excitation fluence, for example by increased trap filling as excitation fluence increases. Such trap filling should cause an apparent increase in mobility with fluence. Whilst it is possible that this is happening in our samples to some extent, the dominant trend we observe is a decrease in the sum of mobilities with excitation density, suggesting that the effects of trap filling are minor. Hence, we present the maximum sum of mobilities calculated over all excitation fluences.

7. In page 12, the statement of "bulk recombination dominates over interfacial recombination at the PTAA/perovskite interface" is not supported by experimental results.

We agree and thank the reviewer for their observation, this statement has now been removed.

Besides, the statement of "a significant reduction in the gradient of the PLQE against excitation intensity of PTAA/perovskite stacks (Figure 2c), which could be due to a significant increase in the background carrier density of the perovskite." is not supported by experimental results neither. This statement is unreasonable because the PLQE of PTAA/perovskite sample is still stronger than that of glass/perovskite sample after 288 h ageing, which cannot be explained by the increased p-type doping.

We have performed PLQE simulations which indicate that an increased, flattened PLQE can be explained by an increase in p-doping in the presence of traps. Details are now provided in Supplementary Note 2:

“Supplementary Note 2: Change in PLQE with Fermi level

To confirm whether the trend we observed in the intensity-dependent PLQE of glass/PTAA/perovskite samples can indeed be explained by an increase in background hole density, we performed PLQE simulations. The system modelled is described in the methods. The Fermi level was varied from 0.05-0.5 eV (from the CB), and the single deep electron trap density was varied from 0 for figure S10a to 10^{15} cm^{-3} for figure S10b. All other parameters were kept constant (Table S2).

We find that a downshift in Fermi level along with the presence of non-radiative recombination, as in figure S10b, can indeed explain the trend we observe in the intensity-dependent PLQE measurements of glass/PTAA/perovskite samples (Figure 2c in the main text).

Figure S10. PLQE of perovskite films, simulated as a function of excitation fluence at a range of Fermi level positions a) with no trap-assisted recombination and b) with trap-assisted recombination.”

8. In page 15, the authors mentioned that “Devices using PTAA are more stable during the first 100 hours of aging”. This description is inappropriate because both kinds of devices (using PTAA or PEDOT:PSS) show significant decay, and the differences in the normalized PCE at 100 hours are more likely in the range of error.

We agree and thank the reviewer for their observation, this statement has now been adjusted to refer to the steady-state J_{sc} loss rather than the PCE.

9. There are some ambiguous descriptions in page 20, between line 493-501. The discussion is difficult to understand, and poorly supported by experimental results.

We thank the reviewer for this feedback. This paragraph has now been completely removed and replaced by justification of the mechanism causing the different degradation behaviour observed in devices with PEDOT:PSS (pg. 19-20).

“The proposed difference in the rate of mobile ion (iodine vacancy) formation in lead-tin perovskites when aged on PEDOT:PSS compared to PTAA can be explained by considering the chemical properties of PEDOT:PSS. Whilst slow iodine vacancy formation is expected in all perovskite materials during aging,⁸ this process can be accelerated by chemical interaction between perovskite and HTL. PSS contains sulfonic acid groups (-SO₃H) which can react with I⁻ from the perovskite to form HI, leaving iodine vacancies in the perovskite film.¹ As both iodine vacancies in the perovskite and protons in PEDOT:PSS are mobile (under illumination and/or heat),^{13,14} this reaction is not limited to the interface but can continue until the PEDOT:PSS or perovskite layers are significantly depleted. HI vapor is expected to be highly mobile in the perovskite¹⁵ and may easily travel to CTLs or surfaces to react further, such that iodine vacancies are permanently left in the film. It is also possible that direct complexation between PEDOT and I⁻ contributes to the production of iodine vacancies in the perovskite.¹⁶ In contrast, PTAA is expected to be more chemically inert and is not expected to accelerate the formation of iodine vacancies in these ways. Hence, the increased ion redistribution-related performance losses observed during aging when using PEDOT:PSS HTLs is consistent with the formation of a higher density of iodine vacancies due to chemical interaction between PSS and the perovskite.”

The input of this study is more incremental rather than disruptive. I don't believe it fits with the standards of Nat. Commun. in current version.

The additional experiments and analysis we have added in the revised version of this paper now reveal a mechanism behind the different degradation observed in devices using PEDOT:PSS and those using PTAA during aging. We now confidently quantify the impact and timescale of four separate degradation modes, namely iodine vacancy formation (dominant when using PEDOT:PSS, over timescales of hours-days), decreased bulk lifetimes,

self-p-doping (dominant when using PTAA, over slightly longer timescales of days-weeks), and CsSnI₃ phase formation (HTL-independent, but very slow over timescales of weeks-months).

This highlights two clear priorities for improving the stability of lead-tin perovskite devices; namely the elimination of acidic PEDOT:PSS, and further suppression of the defect formation leading to increased recombination and p-doping. We believe this constitutes an important addition to the field.

Reviewer #2 (Remarks to the Author):

The manuscript explores the causes of instability of tin-lead perovskite solar cells under combined heat and light stress (ISOS L-2 conditions). It aims to address the questions by disentangling the degradation occurring in various components in the device stack and in fully fabricated solar cells. Findings show that lead-tin perovskite films are stable beyond the usual timescales associated with device degradation. Mobile ions are the major cause of early-time performance degradation, and it can be mitigated by selecting an alternative hole transport layer. However, the authors do not provide strong and sufficient evidence for some findings and statements. Overall, the work needs further major revision before consideration for publication in Nature Communications. Some detailed comments are as follows:

We thank the reviewer for their detailed and specific comments. We have made the requested changes and added evidence where needed. We have also added longer aging time TPC measurements to confirm the rate of change in doping density in the neat perovskite, as well as extensive J-V simulations to support our proposed mechanisms.

1. To illustrate that the new phase peaks are ascribed to the appearance of degradation products, the XRD patterns of fresh ITO/PEDOT: PSS or ITO/PTAA/perovskite samples should be provided to present their differences with aging ones (aged 2160 hrs encaps, 1 sun, 65°C. Figure S2).

Thank you for the observation, these XRD measurements have now been added to figure S2.

2. For cubic perovskite cells fitted to PbSn-perovskite films with same stacks of Glass/Al₂O₃ NPs/ FA_{0.83}Cs_{0.17}Pb_{0.5}Sn_{0.5}I₃, even though they show respective slight expansion of cubic lattice volume after aging under conditions of “unencapsulated ambient, room temperature” and “unencapsulated in N₂, 65°C, 0.76 suns”, the same stacks of fresh samples should have same unit cell volumes, but they show different parameters in Table S1. Please give an explanation.

These measurements were performed on two different samples that were manufactured with an identical procedure, and were found to have unit cell volumes of $250.16 \pm 0.10 \text{ \AA}^3$ and $250.39 \pm 0.10 \text{ \AA}^3$, respectively. The stated uncertainties stem from the uncertainty of the

lattice parameters determined by a Pawley fit to the experimental XRD data, but do not take into account systematic sources of error which may include specimen displacement and misalignment during the XRD measurement. Considering that the cubic lattice parameters yielding these unit volumes are 6.301 Å and 6.303 Å, respectively, we propose that the difference in determined lattice parameters likely stems from small systematic errors.

In contrast, the expansion of the unit cell we observe during aging is much larger than this difference ($\sim 1-4 \text{ \AA}^3$) and is observed consistently for four sets of samples.

3. In 184-186 lines, authors state that “we observe voids filled with smaller crystallites in the active layer of aged devices (Figure 1d), which we don’t observe in fresh devices”, but no direct evidences (e.g., cross-section SEM image of fresh devices, etc) are not presented in the manuscript and supplementary file. Please provide the corresponding results to evidence the claim.

Thank you for the observation, a cross-sectional SEM image of a fresh device is now provided in figure S3.

4. Some descriptions in the manuscript are not well evidenced. For example, the content in 212-213 lines and the responding conclusion state that “although the perovskite’s bulk structure and optical absorption properties remain stable during the first few hundred hours of aging”. From the provided OM and SEM results, we can know that the perovskite in the device changed after 625 h aging. However, the results of OM and SEM proves that perovskite’s bulk structure remaining stable during the first few hundred hours of aging are absent in the manuscript. What is the longest time that the perovskite structure maintains no change? The authors shall provide the corresponding evidences.

OM and SEM images of neat perovskite films at multiple timesteps during aging are provided in figure S3, and these show that sign of aging only become visible on the surface of films after ~ 1000 hours of aging. From the XRD measurements, the perovskite material in devices is expected to degrade slightly differently to the neat perovskite films as some small amount of degradation phases are already observed after 627 hours of aging. However, as the XRD and SEM measurements require removal of encapsulation and destruction of the device sample, these characterizations were not performed at intermediate timesteps. We deem it

likely that regions of degradation phases begin to form immediately upon aging and slowly grow over time. This is supported by the minor decrease in absorbance we observe in neat perovskite films during aging.

Hence, the formation of these phases seems slow enough to only interfere minimally with device performance during the first 300 hours of aging, during which time other factors dominate device degradation (as explored in sections 2.4 and 2.5, which have now been significantly expanded).

5. In 322-323 lines of the manuscript, the authors state “we find that the p_0 and mobility of isolated lead-tin perovskite films on glass are stable over 300 hours of aging”. However, the authors only show the background hole density and mobility for the glass/perovskite samples up to 150 hours (Figure 2). More comprehensive results shall be provided to support their claims.

We thank the reviewer for spotting this mistake in the text. These properties were originally only measured up to 150 hours for glass/perovskite samples.

We have now fabricated and measured an additional batch of glass/perovskite samples, for which we have determined the background hole density and mobility over 500+ hours to enable more substantial comparison to CTL/perovskite samples (Figure 2a, b). We observe that the background hole density in these samples does in fact begin to increase after ~150 hours. These measurements help explain the cause of the degradation observed in lead-tin perovskite solar cells employing PTAA HTLs (see section 2.5).

6. In Figure 2c, the power intensity-dependence of glass/perovskite sample aging 288 h is missing.

These properties were only measured up to 150 hours for glass/perovskite samples. We recognize that this was unclear in the original figure, so the legend and colours have now been adjusted to make this more visible.

Meanwhile, the PLQE of PEDOT:PSS/perovskite films by aging is always higher than fresh. The authors should provide comprehensive results and detailed reasons for this unusual phenomenon.

We thank the reviewer for raising this, which caused us to carefully reevaluate the data from which we calculated these values. We identified a mistake in our calculation of the PLQE of the 0 hour aged PEDOT:PSS/perovskite samples, which was caused by an error in the background subtraction. We carefully reevaluated all other PLQE calculations and found that the error was limited to this single measurement.

After correcting this error, the PLQE of the fresh PEDOT:PSS/perovskite sample is now higher than at all aging times. When recalculating these values, we decided to also apply a stricter signal-to-noise criterion (ratio of 5 rather than 2) to ensure accuracy of the results, especially at low PLQEs. Some of the longer aged PEDOT:PSS/perovskite samples no longer met this criterion and were hence removed from the plot.

7. The dark conductance of glass/PEDOT:PSS/perovskite stack does not show significant changes after 20 h (Figure 2d), but that of glass/PTAA/perovskite stack shows a significant increase of 50 fold in the aging period from 100 hours to 300 hours. Authors infer that the presence of PTAA during aging induces increased p-doping of lead-tin perovskite. How about the sample Glass/perovskite between 100-300 hours of aging?

(See also the response to comment 5.) We agree that longer-time measurements were necessary and have now repeated these measurements with a second batch of films up to 500+ hours. These measurements reveal that in encapsulated neat lead-tin perovskite films, the background charge carrier density begins to increase after 150 hours of aging. This is similar to, if slightly slower than films deposited on PTAA, which may be due to a slight difference in film quality when formed on a different substrate. This discussion has been integrated into section 2.2.

8. In this work, the authors reveal that the early-stage device degradation of lead-tin PSCs is dominated by mobile ions on the device under heat and light stressing, which can be mitigated using PTAA instead of PEDOT:PSS. However, devices using PTAA experience much larger non-mobile-ion related FF and Jsc losses, which result in worse steady-state PCE

after 288 hours of aging. Therefore, according to contents in current manuscript, it is still not good to use PTAA replacing PEDOT:PSS, and it is still not clear that which hole transport layer is a good replacing alternative of PEDOT:PSS, even though the underlying interface problems are pointed. Does authors try and compare other popular HTL materials, such as NiO_x, SAM, to more clearly reveal the degradation problems in tin-lead perovskite devices and offer the good choice of alternative HTL for degradation-mitigation?

The mechanism for the difference between PEDOT:PSS and PTAA which we have been able to propose in the revised version of the paper (see new content in sections 2.4 and 2.5) implies that it is the chemical interaction between the acidic PEDOT:PSS and the perovskite which causes the rapid, mobile-ion-related device performance degradation. We have now added the following to the end of section 2.4 which makes recommendations for future HTLs based on this (pg. 20):

“An increase in the density of iodine vacancies in devices using PEDOT:PSS during aging can likely be mitigated not only by using PTAA, but also by using deprotonated PSS¹⁷ or many other suitable chemically inert HTLs. Indeed, treatment of PEDOT:PSS with NaOH has already been observed to significantly increase the stability of lead-tin perovskite solar cells during aging under illumination.^{18,19}”

We propose that the p-doping observed in devices with PTAA is not related to any changes involving the PTAA, but is rather a process happening in the isolated perovskite which is counteracted by interaction with PEDOT:PSS (see section 2.5, which has been largely rewritten). This implies that changing the HTL will not solve this problem. Rather, it must be addressed by modification of the perovskite absorber, or better protection from external oxidation agents like air or residual solvents. We have added the following discussion to section 2.5 (pg. 22):

“This large increase in background hole density during aging is not likely to be due to chemical interaction between PTAA and the perovskite, which is relatively inert and has been used to fabricate stable neat-Pb perovskite solar cells.²⁰ We instead propose that some process p-dopes the perovskite during aging no matter which HTL is used, but that the previously described chemical interaction between the perovskite and PEDOT:PSS counteracts this p-doping. Significant iodine vacancy formation is expected to n-dope

perovskites, or shift the Fermi level towards vacuum.⁸ Hence, a p-doping process (such as Sn²⁺ vacancy formation) would be counteracted by increased iodine vacancy formation. This mechanism has been previously experimentally observed in CsSnI₃.²¹ Additionally, it is possible that HI produced by the reaction between acidic PEDOT:PSS and the perovskite further acts as a reducing agent to counteract p-doping processes. This explanation is consistent with both the increased mobile ion redistribution-related performance losses observed in devices using PEDOT:PSS, and the increased background carrier density observed when perovskite is aged on PTAA.

This indicates that whilst reducing the rate of iodine vacancy formation in lead-tin perovskite devices will reduce the losses related to the redistribution of mobile ions, doing so will also allow p-doping related losses to increase (as demonstrated in devices using PTAA). Ultimately, both iodine vacancy formation and p-doping must be avoided to achieve required stabilities in lead-tin perovskite solar cells.”

Unfortunately, we have not been able to produce lead-tin solar cells with other commonly used HTLs (such as SAMS) which have a comparable PCE to those fabricated on PEDOT:PSS and PTAA to further confirm the mechanism we proposed. The perovskite quality and the movement of charge carriers through these lower PCE devices is expected to be quite different, which would make it difficult to draw conclusions from their study.

Reviewer #3 (Remarks to the Author):

In this work, Snaith et al studied the degradation behavior of lead-tin based perovskite solar cells, and claim that the oxidation of Sn²⁺ to Sn⁴⁺ is not a big issue. Instead, the ion migration /accumulation is the major reason underlying the device degradation, but such effect can be mitigated by using different charge transporting layers. The authors present comprehensive and persuasive evidences to strongly support their conclusions. While, much better stability of tin-lead mixed perovskite compared to the pristine tin-based counterparts has been widely observed and reported. Therefore, it will be critically important if this work can present some insightful understandings on the origins at atomic levels. Therefore, I would like to consider publication of this work after addressing the following issues.

We thank the reviewer for their comments, and agree that uncovering atomic level understanding of the relevant mechanisms improves the impact of the paper. To achieve this, we have now added extensive simulations to more deeply interpret our variable-rate J-V scans, and carried out longer-time electronic measurements on the neat perovskite material. We believe these additional experiments have greatly strengthened the work and enabled us to establish a mechanistic understanding of the various degradation modes experiences by lead-tin perovskite solar cells. We further detail these changes in our responses below.

1) The pure tine and lead based perovskite films and devices should be also studied to compare the results with the lead/tin mixed perovskite. This will be helpful to understand the working mechanisms of the improved stability.

We thank the reviewer for this suggestion, and agree that explicitly highlighting the difference in the dominant degradation modes between these perovskite compositions is a helpful addition to the work. The degradation of neat lead and neat tin perovskite solar cells have been previously investigated in other works. To highlight the effect of B-site composition on device stability, we have added comparison of our findings to theirs at various points in the manuscript:

On the rate of p-doping, section 2.5 (pg. 22):

“Whilst we have demonstrated that the increase in background hole density p_0 is relatively slow in encapsulated lead-tin perovskite films and devices during aging (compared to devices

exposed to air, or neat Sn perovskites), our results indicate that increased p-doping becomes a performance-limiting factor after ~100 hours of aging under illumination and heat for the devices using a PTAA HTL. Further research is necessary to clarify whether this self-doping process can be ascribed to the slow ingress of oxygen (and can hence be avoided by even better encapsulation), or to some other chemical interaction within the device (for example with residual DMSO^{22,23}). ”

On the effects of mobile ion induced field screening, section 2.4 (pg. 20):

“Given all this, we argue it is the PEDOT:PSS HTL which causes increased ion-induced losses in PbSn perovskite solar cells during aging, and not the Sn-containing perovskite composition. When PTAA is used instead, the fast J-V scans of lead-tin perovskite devices show ~5% absolute performance degradation after 288 hours of aging, which is comparable to neat-lead devices.²⁴ Whilst the effects of mobile ions have been much less investigated in neat-tin perovskites, it is possible that the strong effect of the perovskite’s high background hole density (10^{18} - 10^{20} cm⁻³)²⁵ on the internal distribution of the electric field²⁶ would outweigh effects from redistribution of a smaller density of mobile ions. ”

2) A further step toward understanding the working mechanism is suggested to be presented, e.g. unveiling it at an atomic level.

We thank the reviewer for their suggestion. We have now formulated a mechanism which explains both the increased p-doping observed in devices using PTAA and the increased losses from the aggregation of mobile ions observed in devices using PEDOT:PSS. We further support the proposed mechanism with extensive J-V simulations, which has now been added to section 2.4. Both sections 2.4 and 2.5 have been largely rewritten and now evaluate this mechanism, with the most relevant sections shown:

“The proposed difference in the rate of mobile ion (iodine vacancy) formation in lead-tin perovskites when aged on PEDOT:PSS compared to PTAA can be explained by considering the chemical properties of PEDOT:PSS. Whilst slow iodine vacancy formation is expected in all perovskite materials during aging,⁸ this process can be accelerated by chemical interaction between perovskite and HTL. PSS contains sulfonic acid groups (-SO₃H) which can react with I from the perovskite to form HI, leaving iodine vacancies in the perovskite film.¹ As both iodine vacancies in the perovskite and protons in PEDOT:PSS are mobile

(under illumination and/or heat),^{13,14} this reaction is not limited to the interface but can continue until the PEDOT:PSS or perovskite layers are significantly depleted. HI vapor is expected to be highly mobile in the perovskite¹⁵ and may easily travel to CTLs or surfaces to react further, such that iodine vacancies are permanently left in the film. It is also possible that direct complexation between PEDOT and I contributes to the production of iodine vacancies in the perovskite.¹⁶ In contrast, PTAA is expected to be more chemically inert and is not expected to accelerate the formation of iodine vacancies in these ways. Hence, the increased ion redistribution-related performance losses observed during aging when using PEDOT:PSS HTLs is consistent with the formation of a higher density of iodine vacancies due to chemical interaction between PSS and the perovskite. “ (pg. 20)

“This large increase in background hole density during aging is not likely to be due to chemical interaction between PTAA and the perovskite, which is relatively inert and has been used to fabricate stable neat-Pb perovskite solar cells.²⁰ We instead propose that some process p-dopes the perovskite during aging no matter which HTL is used, but that the previously described chemical interaction between the perovskite and PEDOT:PSS counteracts this p-doping. Significant iodine vacancy formation is expected to n-dope perovskites, or shift the Fermi level towards vacuum.⁸ Hence, a p-doping process (such as Sn²⁺ vacancy formation) would be counteracted by increased iodine vacancy formation. This mechanism has been previously experimentally observed in CsSnI₃.²¹ Additionally, it is possible that HI produced by the reaction between acidic PEDOT:PSS and the perovskite further acts as a reducing agent to counteract p-doping processes. This explanation is consistent with both the increased mobile ion redistribution-related performance losses observed in devices using PEDOT:PSS, and the increased background carrier density observed when perovskite is aged on PTAA.

This indicates that whilst reducing the rate of iodine vacancy formation in lead-tin perovskite devices will reduce the losses related to the redistribution of mobile ions, doing so will also allow p-doping related losses to increase (as demonstrated in devices using PTAA).

Ultimately, both iodine vacancy formation and p-doping must be avoided to achieve required stabilities in lead-tin perovskite solar cells.” (pg. 22)

3) Vertical carrier mobility in a perovskite solar cells is more important compared to the

lateral carrier mobility. However, this work only presents the lateral mobility. Therefore, the carrier mobility along the vertical direction should be examined.

We thank the reviewer for their suggestion and have further investigated whether a change in vertical mobility may impact device performance during aging. Our device simulation already suggested that a decrease in vertical mobility alone would not be sufficient to reproduce the shape of our aged J-V curves, although it may still contribute (Figure 5). To investigate further, we added new analysis of the changes in the EQE under low and high energy illumination during aging for both device architectures (section 2.3, pg. 14):

“We compare the values of the EQE of both devices under high and low wavelength illumination (Figure 3b). For both HTLs, the difference in EQE at the two wavelengths is relatively small, and the EQE at both is significantly degraded after 300 hours of aging. This indicates that J_{sc} degradation is dominated by a decrease in charge collection efficiency at one or both CTL interfaces during aging, rather than bulk lifetime of diffusion of charge carriers.”

4) Buried interface morphology should be studied to correlate the degradation of perovskite film and device.

Full encapsulation of devices during aging makes the investigation of buried interface morphology challenging in our case. Efforts were made to ensure that the fresh neat perovskite films were as similar to the perovskite in devices as possible, namely by making substrate surface qualities as similar as possible. For example, the PTAA used is expected to be sufficiently thick to ‘smooth’ the ITO surface, and Al₂O₃ NPs were used as an interlayer when perovskite was deposited on glass and PTAA to improve wetting. As a result, we do not observe a difference in the fresh PL peak position (now included in Figure S6), nor any significant differences in XRD or SEM measurements of fresh glass/perovskite, ITO/PEDOT:PSS perovskite, and ITO/PTAA/perovskite samples (which have now been added to Figure S2).

We agree with the reviewer that there remains a possible difference in bulk defect density and homogeneity of neat perovskite films on glass and the perovskite in half-stacks/full devices. However, our device J-V simulations suggests that very small-scale changes in ionic point

defect density are responsible for the PCE degradation we observe (section 2.4), which may not be linked to differences in morphology and would likely be difficult to resolve by buried interface characterization.

5) The authors claim the ion migration and accumulation is the key governing the degradation of lead-tin based devices. However, this work lacks some essential evidences to support this point. The forward and reverse J-V scans at different speeds is not persuasive enough.

To further support our claims regarding the effects of mobile ion accumulation on device performance during aging, we have now added J-V scans at a large range of rates (instead of just one fast and one slow scan), and expanded our interpretation of these scans with J-V simulations of devices containing rapid-moving ions. This enabled us to conclude that that in increase in the density of rapid-moving ions was essential to explain the observed trends in the measured J-V data of devices using PEDOT:PSS.

In addition to this, our arguments build on an existing body of work by others regarding the impacts of bias-driven redistribution of mobile ion on the performance of perovskite solar cells. This has previously been extensively investigated by others using variable rate J-V scanning, as well as transient charge extraction measurements.^{2,3,24} Relatively high rapid-moving mobile ion densities have been determined in a range of perovskite compositions, verified by multiple characterization methods.²⁷ Simulations have indicated that the variation of J-V hysteresis with scan rate can be modelled entirely by the inclusion of mobile ion vacancies.²⁸

References

1. Ke, Q. B., Wu, J.-R., Lin, C.-C. & Chang, S. H. Understanding the PEDOT:PSS, PTAA and P3CT-X Hole-Transport-Layer-Based Inverted Perovskite Solar Cells. *Polymers* **14**, 823 (2022).
2. Le Corre, V. M. *et al.* Quantification of Efficiency Losses Due to Mobile Ions in Perovskite Solar Cells via Fast Hysteresis Measurements. *Solar RRL* **6**, 2100772 (2022).
3. Thiesbrummel, J. *et al.* Universal Current Losses in Perovskite Solar Cells Due to Mobile Ions. *Adv. Energy Mater.* **11**, 2101447 (2021).
4. Eames, C. *et al.* Ionic transport in hybrid lead iodide perovskite solar cells. *Nat Commun* **6**, 7497 (2015).
5. Barboni, D. & De Souza, R. A. The thermodynamics and kinetics of iodine vacancies in the hybrid perovskite methylammonium lead iodide. *Energy Environ. Sci.* **11**, 3266–3274 (2018).
6. Tammireddy, S. *et al.* Temperature-Dependent Ionic Conductivity and Properties of Iodine-Related Defects in Metal Halide Perovskites. *ACS Energy Lett.* **7**, 310–319 (2022).
7. Futscher, M. H. *et al.* Quantification of ion migration in CH₃NH₃PbI₃ perovskite solar cells by transient capacitance measurements. *Mater. Horiz.* **6**, 1497–1503 (2019).
8. Bitton, S. & Tessler, N. Perovskite ionics – elucidating degradation mechanisms in perovskite solar cells *via* device modelling and iodine chemistry. *Energy Environ. Sci.* **16**, 2621–2628 (2023).
9. Singh, S., Siliavka, E., Löffler, M. & Vaynzof, Y. Impact of Buried Interface Texture on Compositional Stratification and Ion Migration in Perovskite Solar Cells. *Adv Funct Materials* 2402655 (2024) doi:10.1002/adfm.202402655.
10. Lim, J. *et al.* Long-range charge carrier mobility in metal halide perovskite thin-films and single crystals via transient photo-conductivity. *Nat Commun* **13**, 4201 (2022).

11. Lim, V. J. -Y. *et al.* Air-Degradation Mechanisms in Mixed Lead-Tin Halide Perovskites for Solar Cells. *Advanced Energy Materials* 2200847 (2022)
doi:10.1002/aenm.202200847.
12. Krückemeier, L., Liu, Z., Kirchartz, T. & Rau, U. Quantifying Charge Extraction and Recombination Using the Rise and Decay of the Transient Photovoltage of Perovskite Solar Cells. *Advanced Materials* **35**, 2300872 (2023).
13. Lee, M. M., Teuscher, J., Miyasaka, T., Murakami, T. N. & Snaith, H. J. Efficient Hybrid Solar Cells Based on Meso-Superstructured Organometal Halide Perovskites. *Science* **338**, 643–647 (2012).
14. Senanayak, S. P. *et al.* Charge transport in mixed metal halide perovskite semiconductors. *Nat. Mater.* **22**, 216–224 (2023).
15. Kerner, R. A., Xu, Z., Larson, B. W. & Rand, B. P. The role of halide oxidation in perovskite halide phase separation. *Joule* **5**, 2273–2295 (2021).
16. Lee, S. *et al.* Buried interface modulation via PEDOT:PSS ionic exchange for the Sn-Pb mixed perovskite based solar cells. *Chemical Engineering Journal* **479**, 147587 (2024).
17. Xia, Y., Yan, G. & Lin, J. Review on Tailoring PEDOT:PSS Layer for Improved Device Stability of Perovskite Solar Cells. *Nanomaterials* **11**, 3119 (2021).
18. Wu, D. *et al.* Enhancing the Efficiency and Stability of Tin-Lead Perovskite Solar Cells via Sodium Hydroxide Dedoping of PEDOT:PSS. *Small Methods* 2400302 (2024)
doi:10.1002/smt.202400302.
19. Wang, Q., Chueh, C.-C., Eslamian, M. & Jen, A. K.-Y. Modulation of PEDOT:PSS pH for Efficient Inverted Perovskite Solar Cells with Reduced Potential Loss and Enhanced Stability. *ACS Appl. Mater. Interfaces* **8**, 32068–32076 (2016).
20. Saliba, M. *et al.* Incorporation of rubidium cations into perovskite solar cells improves photovoltaic performance. *Science* **354**, 206–209 (2016).

21. Rajendra Kumar, G., Kim, H.-J., Karupannan, S. & Prabakar, K. Interplay between Iodide and Tin Vacancies in CsSnI₃ Perovskite Solar Cells. *J. Phys. Chem. C* **121**, 16447–16453 (2017).
22. Pascual, J. *et al.* Origin of Sn(II) oxidation in tin halide perovskites. *Mater. Adv.* **1**, 1066–1070 (2020).
23. Sakai, N. *et al.* Adduct-based p-doping of organic semiconductors. *Nat. Mater.* **20**, 1248–1254 (2021).
24. Thiesbrummel, J. *et al.* Ion-induced field screening as a dominant factor in perovskite solar cell operational stability. *Nat Energy* (2024) doi:10.1038/s41560-024-01487-w.
25. Savill, K. J. *et al.* Impact of Tin Fluoride Additive on the Properties of Mixed Tin-Lead Iodide Perovskite Semiconductors. *Adv. Funct. Mater.* **30**, 2005594 (2020).
26. Pena-Camargo, F. *et al.* Revealing the doping density in perovskite solar cells and its impact on device performance. *Applied Physics Reviews* (2022).
27. Diekmann, J. *et al.* Determination of Mobile Ion Densities in Halide Perovskites via Low-Frequency Capacitance and Charge Extraction Techniques. *J. Phys. Chem. Lett.* **14**, 4200–4210 (2023).
28. Courtier, N. E., Cave, J. M., Foster, J. M., Walker, A. B. & Richardson, G. How transport layer properties affect perovskite solar cell performance: insights from a coupled charge transport/ion migration model. *Energy Environ. Sci.* **12**, 396–409 (2019).

Response to reviewers

Reviewer #1 (Remarks to the Author):

In this revised manuscript, the decay processes of Pb-Sn perovskites on PEDOT:PSS and PTAA have been analyzed and described more appropriately. After modification, the contribution of this study has been better clarified. What convinces and attracts me most is their reformulated statement of: the chemical interaction between the perovskite and PEDOT:PSS counteracts the detrimental p-doping effect that was observed in PTAA/Perovskite case. Since this idea, the rest of the discussion becomes innovative and reasonable. The authors have collected a lot of data to compare the different decay processes between “PEDOT:PSS /Perovskite” and “PTAA/Perovskite”, but lack a clear idea in their first version of manuscript. By admitting that “although lead-tin PSCs employing PTAA are less affected by mobile ion redistribution-related losses during aging, an increase in p-doping during aging leads to even worse overall degradation”, in my opinion, the authors are accessing a more attractive conclusion, i.e., the oxidation of Sn in the “PTAA/Perovskite” case is more obvious than that in the case of “PEDOT:PSS /Perovskite”, and this oxidation can be weakened by the presence of iodine vacancies induced by PEDOT:PSS. So, at this point, it is reasonable to speculate that the oxidation of Sn might be related to the dynamically generated, reactive iodine.

This revised manuscript can be accepted, but it is lengthy and difficult to read. I suggest the authors try their best to reorganize the writing.

We are pleased to hear the reviewers' positive evaluation of (and engagement with) our added analysis, and thank them for their time. We have restructured/rewritten some of the paper to attempt to make it easier to read and understand. Most notably, we have:

- moved part of section 2 (“Opto-electronic changes during aging”) to the methods
- reorganized and slightly shortened section 4 (“Impact of mobile ion redistribution on device performance during aging”)

- added a diagram illustrating the final proposed mechanism of performance degradation (Figure 6)

Reviewer #2 (Remarks to the Author):

This work studies the degradation occurring in various components in the device stack and fully fabricated solar cells. Findings show that lead-tin perovskite films are stable beyond the usual timescales associated with device degradation. Mobile ions are the major cause of early-time performance degradation, and it can be mitigated by selecting an alternative hole transport layer. The authors have provided evidence to support the findings and statements. The work can be considered for publication in Nature Communications after addressing the following issues.

We thank the reviewer for their time and their useful suggestions to improve the work further.

1. For the statement “we observe voids filled with smaller crystallites in the active layer of aged devices (Figure 1d), which we don’t observe in fresh devices”, the authors have added the evidence of cross-section SEM image of fresh devices in Figure S3. It is suggested to merge Figure S3 into Figure 1d to show the comparison of fresh and aged C-SEM images.

This has now been done.

2. The authors answered that XRD and SEM measurements were not performed at intermediate timesteps due to the difficulty of removal of encapsulation and destruction of the device. However, if the exact timeline of perovskite structure change is uncertain, the authors should revise the expression according to their experimental results. The authors shall add wording (e.g., it is a hypothesis that ...) in the statements without experimental proofs.

We believe that we have now corrected this:

“Although it is possible that some growth of small regions of δ CsSnI₃ may begin immediately during aging, such growth seems sufficiently slow for the perovskite bulk to be left largely intact during 600+ hours of device aging. The measurements lead us to hypothesize that the evolution

of the degradation products is relatively slow, and the original lead-tin perovskite phase remains stable over long periods of stressing under elevated temperatures and illumination, even in half-stacks with HTLs or in complete devices.”

3. In Figure 2a, the annotations of blue and purple lines are the same (“glass/perovskite”). Please check and correct them.

We have now fixed this.

4. There are some format errors in the text, like “288 hours”, and reference format errors. Please carefully check and correct.

We believe that these have now been corrected.

Reviewer #3 (Remarks to the Author):

The paper was well revised, and I suggest the acceptance of it for publication.

We are pleased to read the reviewer’s favorable evaluation and thank them for their time.